# A Classification of $G$-Invariant Shallow Neural Networks

**Devanshu Agrawal & James Ostrowski**
Department of Industrial and Systems Engineering
University of Tennessee
Knoxville, TN 37996
dagrawa2@vols.utk.edu, jostrows@utk.edu

## Abstract

When trying to fit a deep neural network (DNN) to a $G$-invariant target function with $G$ a group, it only makes sense to constrain the DNN to be $G$-invariant as well. However, there can be many different ways to do this, thus raising the problem of "$G$-invariant neural architecture design": What is the optimal $G$-invariant architecture for a given problem? Before we can consider the optimization problem itself, we must understand the search space, the architectures in it, and how they relate to one another. In this paper, we take a first step towards this goal; we prove a theorem that gives a classification of all $G$-invariant single-hidden-layer or "shallow" neural network ($G$-SNN) architectures with ReLU activation for any finite orthogonal group $G$, and we prove a second theorem that characterizes the inclusion maps or "network morphisms" between the architectures that can be leveraged during neural architecture search (NAS). The proof is based on a correspondence of every $G$-SNN to a signed permutation representation of $G$ acting on the hidden neurons; the classification is equivalently given in terms of the first cohomology classes of $G$, thus admitting a topological interpretation. The $G$-SNN architectures corresponding to nontrivial cohomology classes have, to our knowledge, never been explicitly identified in the literature previously. Using a code implementation, we enumerate the $G$-SNN architectures for some example groups $G$ and visualize their structure. Finally, we prove that architectures corresponding to inequivalent cohomology classes coincide in function space only when their weight matrices are zero, and we discuss the implications of this for NAS.

## 1 Introduction

When trying to fit a deep neural network (DNN) to a target function that is known to be $G$-invariant with respect to a group $G$, it is desirable to enforce $G$-invariance on the DNN as prior knowledge. This is a common scenario in many applications such as computer vision, where the class of an object in an image may be independent of its orientation [Veeling et al., 2018], or point clouds that are permutation-invariant [Qi et al., 2017]. Numerous $G$-invariant DNN architectures have been proposed over the years, including $G$-equivariant convolutional neural networks ($G$-CNNs) [Cohen and Welling, 2016], $G$-equivariant graph neural networks [Maron et al., 2019a], and a DNN stacked on a $G$-invariant sum-product layer [Kicki et al., 2020]. However, it is unclear which of these architectures a practitioner should choose for a given problem, and even after one is selected, additional design choices must be made; for $G$-CNNs alone, the practitioner must select a sequence of representations of $G$ to determine the composition of layers, and it is unknown how best to do this. Moreover, despite a complete classification of $G$-CNNs [Kondor and Trivedi, 2018, Cohen et al., 2019b], it is unknown if every $G$-invariant DNN is a $G$-CNN, and hence the "optimal" $G$-invariant architecture may not even exist in the space of $G$-CNNs.

36th Conference on Neural Information Processing Systems (NeurIPS 2022).

For some architectures, universality theorems exist guaranteeing the approximation of any $G$-invariant function with arbitrarily small error [Maron et al., 2019b, Ravanbakhsh, 2020, Kicki et al., 2020], and it is thus tempting to conclude that these universal architectures are sufficient for all $G$-invariant problems. However, it is well-known that universality [Cybenko, 1989] alone is not a sufficient condition for a good DNN model and that the function subspaces that a network traverses as it grows to the universality limit is just as important as the limit itself. This suggests that the way in which a DNN is constrained to be $G$-invariant does matter, and different $G$-invariant architectures may be suitable for different problems. This raises the fundamental question: For a given problem, what is the "best" way to constrain the parameters of a DNN such that it is $G$-invariant?

This paper takes a first step towards answering the above question. Specifically, before we can consider the optimization problem for the best $G$-invariant architecture, we must understand the search space: What are all the possible ways to constrain the parameters of a DNN such that it is $G$-invariant, and how are these different $G$-invariant architectures related to one another?

The above is a special case of the broader and more fundamental problem of neural architecture design. One of the most prominent approaches to this problem in the literature is neural architecture search (NAS), which at its core is trial-and-error [Elsken et al., 2019]. While trial-and-error is—in principle—straightforward for determining, e.g., the optimal depth or hidden widths of a DNN, it is less clear for $G$-invariant architectures, where a practitioner does not even know all their options. More generally, NAS presupposes knowledge about which architectures are in the search space, which ones are not, which ones are equivalent or special cases of others, and how best one should move from one architecture to another. Thus, to apply even the simplest approach to $G$-invariant neural architecture design, we must first be able to enumerate all $G$-invariant architectures.

Our main result is Thm. 4, which gives a classification of all $G$-invariant single-hidden-layer or "shallow" neural network ($G$-SNN) architectures with rectified linear unit (reLU) activation, for any finite orthogonal group $G$ acting on the input space. More precisely, every $G$-SNN architecture can be decomposed into a sum of "irreducible" ones, and Thm. 4 classifies these. The classification is based on a correspondence of each irreducible architecture to a representation of G in terms of its action on the hidden neurons via so-called "signed permutations", where the representation is required to satisfy an additional condition to eliminate degenerate (linear) architectures and redundant architectures equivalent to simpler ones. The classification then boils down to the classification of these representations. These representations, and hence the corresponding architectures as well, are classified in terms of the first cohomology classes of $G$ and thus admit a topological interpretation. We note that, while connections between neural networks and the group of signed permutations have been previously made in the literature [Ojha, 2000, Negrinho and Martins, 2014, Arjevani and Field, 2020], to our knowledge, no such connection has yet been leveraged to begin a classification program of $G$-invariant architectures.

We also prove Thm. 5, which characterizes the "network morphisms" linking irreducible $G$-SNN architectures in architecture space. In NAS, network morphisms furnish a topology on architecture space and describe how one should move from one architecture to another during the search [Wei et al., 2016]. Taken together, Thms. 4&5 give a complete description of $G$-SNN architecture space.

This paper is perhaps most similar in spirit to the works of Kondor and Trivedi [2018] and Cohen et al. [2019b] and draws on similar mathematical machinery; like them, this paper's contribution is also primarily theoretical. Kondor and Trivedi [2018] prove that every $G$-invariant DNN is a $G$-CNN under the assumption that every affine layer is $G$-equivariant; that this is true without the assumption is only conjectured. Cohen et al. [2019b] generalize this to $G$-CNNs where hidden activations are vector fields and provide a classification of all $G$-CNNs, but the conjecture of Kondor and Trivedi [2018] is left open. In contrast to these works, in our paper, we do not assume the pre-activation affine transformation to be $G$-equivariant– only that the whole network is $G$-invariant. Thus, a future extension of Thm. 4 to deep architectures would either prove or refute the cited conjecture, at least for ReLU networks. Moreover, these works do not explicitly work out the group representations compatible with ReLU, and other works [Cohen and Welling, 2016, Cohen et al., 2019a] consider only unsigned permutations with ReLU. In contrast, our classification reveals $G$-SNN architectures (namely, those corresponding to proper signed permutation representations or nontrivial cohomology classes) that, to our knowledge, have never been explicitly identified in the literature previously.

We also note the work of Maron et al. [2019a], who classify all $G$-equivariant linear layers for graph neural networks; however, they restrict their attention to unsigned permutation representations only, and there classification is again not guaranteed to contain all $G$-invariant ReLU networks.

The remainder of the paper is organized as follows: In Sec. 2, we give a classification of the "signed permutation representations" of $G$ and relate these representations to the cohomology classes of $G$.[1] Then in Sec. 3, we build towards and state our main classification theorem of $G$-SNN architectures. While Sec. 2 makes little reference to $G$-SNNs, presenting it upfront helps to streamline the exposition in Sec. 3, with much of the notation and terminology established. In Sec. 4, we visualize the $G$-SNN architectures for some example groups $G$, and in Sec. 5, we make a number of remarks including a theorem on the "network morphisms" between $G$-SNN architectures. Finally, in Sec. 6, we end with conclusions and next steps towards the problem of $G$-invariant neural architecture design.

## 2 Signed permutation representations

### 2.1 Preliminaries

Throughout this paper, let $G$ be a finite group of $m \times m$ orthogonal matrices. Let $\mathcal{P}(n)$ be the group of all $n \times n$ permutation matrices and $\mathcal{Z}(n)$ the group of all $n \times n$ diagonal matrices with diagonal entries $\pm 1$. Let $\mathrm{PZ}(n) = \mathcal{P}(n) \ltimes \mathcal{Z}(n)$, which is the group of all *signed permutations*– i.e., the group of all permutations and reflections of the standard orthonormal basis $\{e_1, \ldots, e_n\}$. This group is also called the hyperoctahedral group in the literature [Baake, 1984].

A *signed permutation representation* (signed perm-rep) of degree $n$ of $G$ is a homomorphism $\rho : G \mapsto \mathrm{PZ}(n)$. Whenever we say $\rho$ is a signed perm-rep, let it be understood that its degree is $n$ unless we say otherwise. A signed perm-rep $\rho$ is said to be *irreducible* if for every $i, j = 1, \ldots, n$, there exists $g \in G$ such that $\rho(g)e_i = \pm e_j$. As we will see in Sec. 3.2, every $G$-SNN can be written as a sum of "irreducible" $G$-SNNs, and every irreducible $G$-SNN corresponds to an irreducible signed perm-rep. It is therefore sufficient for our purposes to classify all irreducible signed perm-reps of $G$; moreover, this need only be done up to conjugacy as seen next.

### 2.2 Classification up to conjugacy

Two signed perm-reps $\rho, \rho'$ are said to be *conjugate* if there exists $A \in \mathrm{PZ}(n)$ such that $\rho'(g) = A^{-1}\rho(g)A \forall g \in G$. We let $\rho^{\mathrm{PZ}}$ denote the conjugacy class of the signed perm-rep $\rho$. Note that conjugation preserves the (ir)reducibility of a signed perm-rep (Prop. 7 in Supp. A.1), and it thus makes sense to speak of the irreducibility of an entire conjugacy class $\rho^{\mathrm{PZ}}$. The significance of the conjugacy relation is that conjugate signed perm-reps correspond to the same $G$-SNN (see Sec. 3.2); we are thus interested in the classification of irreducible signed perm-reps only up to conjugacy.

Our first theorem below gives the desired classification of signed perm-reps.[2] For $H, K \leq G$, let $(H, K)^G$ denote the *paired conjugacy class*

$$(H, K)^G = \{(g^{-1}Hg, g^{-1}Kg) : g \in G\}.$$

Define the following set of conjugacy classes of subgroup pairs:

$$\mathcal{C}^G_{\leq 2} = \{(H, K)^G : K \leq H \leq G \mid |H : K| \leq 2\}.$$

For every $(H, K)^G \in \mathcal{C}^G_{\leq 2}$, we define a signed perm-rep $\rho_{HK}$ as follows: Let $(g_1, \ldots, g_n)$ be a transversal of $G/H$ with $g_1 \in K$. For each $i = 1, \ldots, n$, define $g_{-i} = g_i h$ for some $h \in H \setminus K$ if $|H : K| = 2$ and $h = 1$ if $|H : K| = 1$. Then define the signed perm-rep $\rho_{HK}$ such that $\rho_{HK}(g)e_i = e_j$ if $gg_iK = g_jK$ for every $i, j = \pm 1, \ldots, \pm n$.

**Theorem 1.** *We have:*

  *(a) Every $\rho_{HK}$ is irreducible.*

---

[1]We assume some familiarity with group theory including semidirect products and quotient groups, conjugacy classes of subgroups, and group action [see Herstein, 2006].

[2]All proofs, as well as additional lemmas and useful propositions, can be found in the supplementary material.

*(b) For every irreducible signed perm-rep $\rho'$, there exists a unique $(H, K)^G$ such that $\rho'$ is conjugate to $\rho_{HK}$.*

Theorem 1 equivalently states that the set

$$\mathcal{R}^{\mathrm{PZ}}(G) = \{\rho_{HK}^{\mathrm{PZ}} : (H, K)^G \in \mathcal{C}_{\leq 2}^G\}$$

is a partition on the set of all irreducible signed perm-reps into conjugacy classes. We will say $\rho_{HK}$ has *type* $|H : K|$—i.e., type 1 if $H = K$ and type 2 otherwise. For the interested reader, we note that $\rho_{HK}$ is the rep of $G$ induced from the rep $\phi : H \mapsto \{-1, 1\}$ where $K = \ker(\phi)$.

### 2.3 Group cohomology

Group cohomology offers an alternative perspective on the classification of signed perm-reps and is the basis for the directed graph visualizations in Sec. 4.1 and Supp. C.2.[3] Note, however, that a technical understanding of group cohomology is not required for most of this paper, and we give only a high-level overview here. Let $\rho$ be a signed perm-rep and $\pi : G \mapsto \mathcal{P}(n)$ and $\omega : G \mapsto \tilde{\mathcal{Z}}(n)$ the unique functions satisfying $\rho(g) = \omega(g)\pi(g) \forall g \in G$. The function $\omega$ is called a *cocycle* and describes the sign flips associated to the action of $G$ through $\rho$. It can be depicted using colored directed graphs as in Fig. 1 (and Figs. 4-6), where arcs of different colors represent the actions of diffrent generators of $G$ and dashed arcs represent sign flips. The cocycle $\omega$ thus encodes topological information about how a space "twists" as we move through it by the action of $G$.

If $\rho'$ is another signed perm-rep conjugate to $\rho$ by a diagonal matrix in $\mathcal{Z}(n)$, then its corresponding cocycle $\omega'$ is said to be *cohomologous* to $\omega$, and the set of all cocycles cohomologous to $\omega$ is said to form a *cohomology class*. the reason for this equivalence is that the number of sign flips around a cycle is unique only up to an even number of sign flips; thus, the solid vs. dashed arcs of the directed graphs in Fig. 1 are not unique. The set of all cohomology classes associated to a single $\pi$ forms a *cohomology group*, and it turns out that the classification of these cohomology groups and their elements gives another way to look at the classification of the signed perm-reps of $G$; see Prop. 15 in Supp. A.3 for details. Type 1 signed perm-reps are then the ones corresponding to the identity elements of these cohomology groups—i.e., the cohomology classes with no sign flips—and type 2 signed perm-reps correspond to the elements describing nontrivial sign flip patterns.

Finally, as we are only interested in signed perm-reps up to conjugation, we regard two cohomology classes as equivalent if their corresponding signed perm-reps are related by conjugation with a permutation matrix in $\mathcal{P}(n)$.[4] Concretely, this ensures that colored directed graphs isomorphic to those shown in Fig. 1 are in fact considered equivalent to them.

## 3 Classification of $G$-SNNs

### 3.1 Canonical parameterization

A *shallow neural network* (SNN) is a function $f : \mathbb{R}^m \mapsto \mathbb{R}$ of the form

$$f(x) = a^\top \mathrm{ReLU}(Wx + b) + d, \tag{1}$$

where $W \in \mathbb{R}^{n \times m}$ and $a, b \in \mathbb{R}^n$ for some $n$, and $d \in \mathbb{R}$. Here ReLU is the *rectified linear unit* activation function defined as $\mathrm{ReLU}(x) = \max(0, x)$ elementwise. The parameterization of an SNN given in Eq. 1 contains redundancies in the sense that different parameter configurations can define the same function. For example, applying a permutation to the rows of $a$, $b$, and $W$ generally results in a different parameter configuration but always leaves $f$ invariant. Also, by the identity

$$\mathrm{ReLU}(zx) = \mathrm{ReLU}(x) - H(-z)x, x \in \mathbb{R}, z \in \{-1, 1\}, \tag{2}$$

(where $H$ is the Heaviside step function; see Prop. 16 in Supp. B), the reflection of one or more rows of the augmented matrix $[W \mid b]$ in Eq. 1 together with the addition of a linear term—which can be represented as the sum of two hidden neurons—leaves $f$ invariant; e.g.,

$$f(x) = a^\top \mathrm{ReLU}[-(Wx + b)] + \mathrm{ReLU}[a^\top(Wx + b)] - \mathrm{ReLU}[-a^\top(Wx + b)] + d.$$

---

[3]These visualizations are based on a geometric perspective of cohomology on Cayley graphs [Druțu and Kapovich, 2018, sec. 5.9]; see [Tao, 2012] for intuition.

[4]For the interested reader, this equivalence amounts to quotienting the cohomology group $\mathcal{H}^1(G, M)$ by the automorphism group of the $G$-module $(M, \pi)$; see Prop. 15 (b) in Sup. A.3 for details.

Note that these permutation and reflection redundancies form the group $\mathrm{PZ}(n)$. The lemma below will help us define a "canonical parameterization" in which such redundancies are eliminated.

**Lemma 2.** *Let $\Theta_n$ be the set of all augmented matrices $[W \mid b] \in \mathbb{R}^{n \times (m+1)}$ such that the rows of $W$ have unit norms and no two rows of $[W \mid b]$ are parallel. Let $\Omega_n \subset \Theta_n$ be a fundamental domain[5] under the action of $\mathrm{PZ}(n)$. Let $f : \mathbb{R}^m \mapsto \mathbb{R}$ be an SNN of the form in Eq. 1. Then there exist unique $n_* \in \mathbb{N}$, $[W_* \mid b_*] \in \Omega_{n_*}$, $a_* \in \mathbb{R}^{n_*}$ with nonzero elements, $c_* \in \mathbb{R}^m$, and $d_* \in \mathbb{R}$ such that:*
$$f(x) = a_*^\top \mathrm{ReLU}(W_* x + b_*) + c_*^\top x + d_* \forall x \in \mathbb{R}^m.$$

Here, since the rows of $[W_* \mid b_*]$ are pairwise nonparallel, then no two hidden neurons can form a linear term; all such hidden neurons are collected in the unique term $c_*^\top x$. As a result, $n_* \leq n$ as $n_*$ is the smallest number of hidden neurons possible. We refer to the unique parameterization of the SNN $f$ in terms of $(a_*, b_*, c_*, d_*, W_*)$ as its *canonical parameterization*, and the SNN is then said to be in *canonical form*. We call $b_*$, $W_*$, and the rows of $W_*$ the canonical *bias*, *weight matrix*, and *weight vectors* of the $G$-SNN respectively. Note that the canonical parameterization is a function of the choice of fundamental domain $\Omega_{n_*}$.

### 3.2 $G$-SNNs and signed perm-reps

For $f$ to be a $G$-invariant SNN ($G$-SNN), the action of $g \in G$ on the domain of $f$ must be equivalent to one of the redundancies in the parameterization of SNNs. In the canonical parameterization, however, this means that the parameters of $f$ and of $f \circ g$ must be identical. This places constraints on the canonical parameters of a $G$-SNN, as made precise in the lemma below.

**Lemma 3.** *Let $f : \mathbb{R}^m \mapsto \mathbb{R}$ be an SNN expressed in canonical form with respect to a fundamental domain $\Omega_{n_*} \subset \Theta_{n_*}$. Then $f$ is $G$-invariant if and only if there exists a unique signed perm-rep $\rho : G \mapsto \mathrm{PZ}(n_*)$ such that the canonical parameters of $f$ satisfy the following equations for all $g \in G$:*

$$\rho(g)W_* = W_* g \tag{3}$$
$$\pi(g)a_* = a_* \tag{4}$$
$$\rho(g)b_* = b_* \tag{5}$$
$$gc_* = c_* + \frac{1}{2}(I - g)W_*^\top a_*. \tag{6}$$

We see that the constraints on the canonical parameters of a $G$-SNN are not necessarily unique, as they depend on a signed perm-rep $\rho$, whence the classification program of $G$-SNNs in this paper. Lemma 3 thus establishes the promised connection between $G$-SNNs and signed perm-reps. To formalize this correspondence, let $\mathrm{SNN}(G)$ be the set of all $G$-SNNs and $\mathcal{R}^{\mathrm{PZ}}(G)$ the set of all conjugacy classes of signed perm-reps of $G$. Define the map $F : \mathrm{SNN}(G) \mapsto \mathcal{R}^{\mathrm{PZ}}(G)$ such that if $f \in \mathrm{SNN}(G)$ and $\rho$ is the corresponding signed perm-rep appearing in Lemma 3, then $F(f) = \rho^{\mathrm{PZ}}$. Then $F$ is a well-defined function in the sense that $F(f)$ does not depend on the choice of fundamental domain $\Omega_{n_*}$. Indeed, a change of fundamental domain $\Omega_{n_*} \to \Omega'_{n_*}$ induces a transformation $[W_* \mid b_*] \to A[W_* \mid b_*]$ for a unique $A \in \mathrm{PZ}(n_*)$. By Eq. 3, this in turn induces the conjugation $\rho \to A\rho(\cdot)A^{-1}$, thereby leaving $F(f) = \rho^{\mathrm{PZ}}$ invariant.

We now define a *$G$-SNN architecture* to be a subset $S \subseteq \mathrm{SNN}(G)$ such that $S = F^{-1}(\rho^{\mathrm{PZ}})$ for some $\rho^{\mathrm{PZ}} \in \mathrm{ran}(F)$; it consists of all $G$-SNNs that are constrained to respect the same representation of $G$. In this language, the purpose of this paper is to classify all $G$-SNN architectures.

A $G$-SNN $f$ is said to be *irreducible* if $F(f)$ is a conjugacy class of irreducible signed perm-reps. Let $f^{\mathrm{PZ}}$ denote the $G$-SNN architecture containing the $G$-SNN $f$, and observe that if $f$ is irreducible, then so are all $G$-SNNs in $f^{\mathrm{PZ}}$; in this case, $f^{\mathrm{PZ}}$ is said to be an *irreducible architecture*.

It can be shown that every $G$-SNN admits a decomposition into a sum of irreducible $G$-SNNs (Prop. 19 in Supp. B.2). It follows that to classify all $G$-SNN architectures, it is enough to classify all irreducible $G$-SNN architectures. This amounts to two tasks: (1) Classify all irreducible signed perm-rep conjugacy classes in $\mathrm{ran}(F)$, and (2) for every irreducible $\rho^{\mathrm{PZ}} \in \mathrm{ran}(F)$, give a parameterization of all $G$-SNNs in the architecture $F^{-1}(\rho^{\mathrm{PZ}})$.

---

[5]If a group $G$ acts on a set $\mathcal{X}$, then a fundamental domain is a set $\Omega \subseteq \mathcal{X}$ such that $\{g\Omega : g \in G\}$ is a partition of $\mathcal{X}$. An example fundamental domain in $\Theta_n$ under the action of $\mathrm{PZ}(n)$ is given in Prop. 18 (see Supp. B.1).

### 3.3 The classification theorem

We now state our main theorem, but first we introduce some notation. If $A$ is a linear operator (resp. set of linear operators), then let $P_A$ be the orthogonal projection operator onto the vector subspace that is pointwise-invariant under the action of $A$ (resp. all elements of $A$). Note that if $A$ is a finite orthogonal group, then [Serre, 1977, sec. 2.6]

$$P_A = \frac{1}{|A|} \sum_{a \in A} a.$$

Let $\mathrm{st}_G(P_A)$ denote the stabilizer subgroup

$$\mathrm{st}_G(P_A) = \{g \in G : gP_A = P_A\}.$$

**Theorem 4.** *Let $\rho_{HK}$ be an irreducible signed perm-rep of $G$, and let $\{g_1, \ldots, g_n\}$ be a transversal of $G/H$ such that $\rho_{HK}(g_i)e_1 = e_i$. Let $\tau = |H : K| - 1$. Then:*

*(a) $\rho_{HK}^{\mathrm{PZ}} \in \mathrm{ran}(F)$ if and only if $\mathrm{st}_G(P_K - \tau P_H) = K$.*

*(b) If $\rho_{HK}^{\mathrm{PZ}} \in \mathrm{ran}(F)$, then $f \in F^{-1}(\rho_{HK}^{\mathrm{PZ}})$ if and only if the canonical parameters[6] of $f$ have the following forms:*

$$W_* = \sum_{i=1}^{n} e_i(g_i w)^\top, w \in \mathrm{ran}(P_K - \tau P_H), \|w\| = 1 \tag{7}$$

$$a_* = a\vec{1}, a \neq 0 \tag{8}$$

$$b_* = (1 - \tau)b\vec{1}, b \in \mathbb{R} \tag{9}$$

$$c_* = -\frac{1}{2}\tau W_*^\top a_* + c, c \in \mathrm{ran}(P_G). \tag{10}$$

The condition in Thm. 4 (a) helps to exclude architectures where Eq. 7 yields redundant weight vectors. Combining this with the classification of irreducible signed perm-reps up to conjugacy (Thm. 1), we obtain a complete classification of the irreducible $G$-SNN architectures as an immediate corollary. Theorems 1&4 can be assembled into an algorithm that enumerates all irreducible $G$-SNN architectures for any given finite orthogonal group $G$. We implemented the enumeration algorithm using a combination of GAP[7] and Python; our implementation currently supports, in principle, all finite permutation groups $G < \mathcal{P}(m)$.[8] Using our code implementation, we enumerated all irreducible $G$-SNN architectures for one permutation representation of every group $G$, $|G| \leq 8$, up to isomorphism. We report the number of architectures, broken down by type, for each group in Table 1 (see Supp. C.1; a discussion is included there as well). A key observation is that the number of type 2 architectures—which to our knowledge have never appeared in the literature previously—is significant; e.g., for the dihedral permutation group $G = D_4$ on four elements, there are five type 1 architectures compared to seven type 2 architectures.

Script execution time for each group was under 2 seconds. Nevertheless, we remark that our code is not optimized for speed and scalability as our purpose was exploration and intuition. As future work, we will work out Thm. 4 for specific families of groups to derive more direct and efficient implementations. We will also investigate how to build $G$-SNN architectures from smaller $G_1$-SNN and $G_2$-SNN architectures where $G$ is a (semi)direct product of $G_1$ and $G_2$. Finally, we note that for the application of NAS, we will probably never enumerate *all* $G$-SNN architectures; instead, we will generate them on-the-fly as we move through the search space.

---

[6]This is an important subtlety. By specifying "canonical parameters", we exclude from Eqs. 7-10 parameter values that do not correspond to a canonical parameterization. For example, $w$ cannot lie in a proper subspace $\mathrm{ran}(P_{K'} - \tau P_{H'}) \subset \mathrm{ran}(P_K - \tau P_H)$ as this would result in at least two rows of $[W_* \mid b_*]$ being parallel. For the same reason, for type 1 architectures, we must have $b \neq 0$ if any two rows of $W_*$ are antiparallel.

[7]GAP is a computer algebra system for computational discrete algebra with particular emphasis on computational group theory [GAP].

[8]Code for our implementation and for reproducing all results in this paper is available at: `https://github.com/dagrawa2/gsnn_classification_code`.

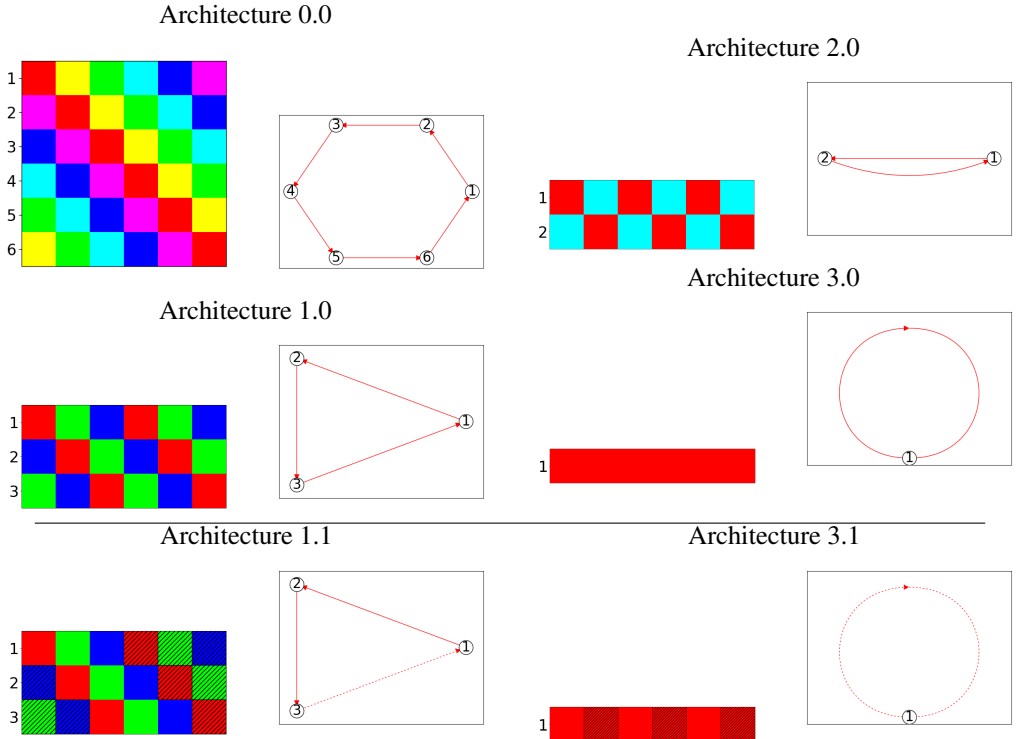

Figure 1: Constraint pattern of the weight matrix and illustration of the cohomology class of each irreducible $G$-SNN architecture for the cyclic permutation group $G = C_6$. The number of rows (resp. columns) in each pattern is the number of hidden (resp. input) neurons in the architecture. In each pattern, weights of the same color and texture (solid vs. hatched) are constrained to be equal; weights of the same color but different texture are constrained to be opposites (colors should not be compared across different architectures). In each cohomology class illustration, the nodes represent the hidden neurons of the architecture, and the arcs represent the action of the generators of $G$ on the rows of the weight matrix (all arcs are the same color because $C_6$ has only one generator). Solid (resp. dashed) arcs preserve (resp. reverse) orientation. See Supp. C.2 for a richer example– the dihedral permutation group $D_6$.

## 4 Examples

### 4.1 The cyclic permutation group

Consider the group $G = C_6$ of all cyclic permutations on the dimensions of the input space $\mathbb{R}^6$.[9] There are six irreducible $G$-SNN architectures for $G = C_6$ (Fig. 1); "architecture i.j" refers to $F^{-1}(\rho^{\mathrm{PZ}}_{H_i K_j})$ where $H_0, \ldots, H_3$ are isomorphic to $C_1$, $C_2$, $C_3$, and $C_6$ respectively and $K_j \leq H_i$ such that $|H_i : K_j| = j + 1$. Architectures i.0 are thus exactly the type 1 ones, and two architectures i.j and i.k for distinct j and k correspond to inequivalent cohomology classes in the same cohomology group. Note that the architectures with $n$ hidden neurons correspond to $H_i \cong C_{\frac{6}{n}}$.

The type 1 architectures i.0 correspond to ordinary unsigned perm reps of $G$. These are the "obvious" architectures that practitioners probably could have intuited. From Fig. 1, we see that the weight matrices of these architectures are constrained to have a circulant structure; cycling the input neurons is thus equivalent to cycling the hidden neurons, leaving the output invariant as all weights in the second layer (not depicted) are constrained to be equal. This circulant structure is also apparent in the cohomology class illustrations.

Architectures i.1 are type 2 and are perhaps less obvious. Cycling the input neurons is equivalent to cycling the hidden neurons only up to sign; if we cycle a weight vector around all the hidden neurons,

<hr />

[9]See Supp. C for richer examples that could not fit in the main paper.

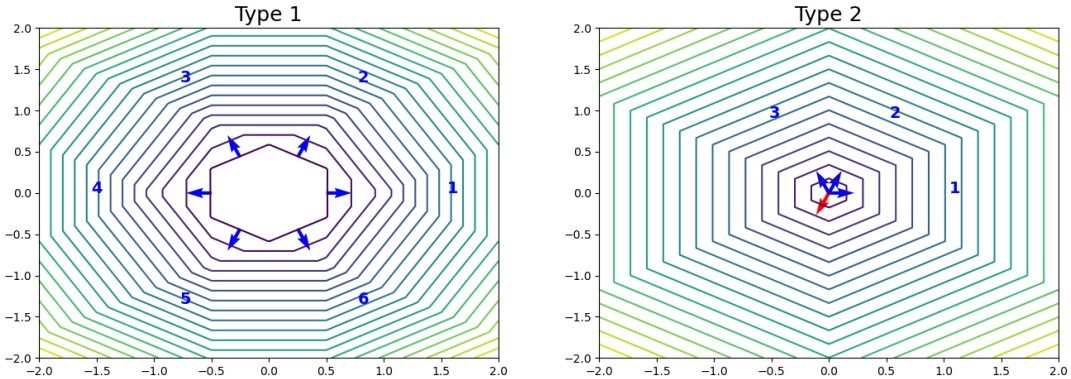

Figure 2: Contour plots of the two irreducible $G$-SNN architectures for the 2D orthogonal representation of $G = C_6$. The blue vectors are the weight vectors– i.e., rows of the weight matrix $W_*$, and their offsets from the origin in the type 1 architecture indicate the bias $b_*$. The red vector in the type 2 architecture is the canonical parameter $c_*$ of the $G$-SNN. See Supp. C.3 for a richer example– the dihedral rotation group $D_6$.

then we do not return to the original weight vector but instead to its opposite. If we think of the dashed arcs in the cohomology class illustrations as "half-twists" in a cylindrical band, then Architectures i.1 correspond to a Möbius band, thereby distinguishing them topologically from architectures i.0. Alternatively, in terms of graph colorings, if the nodes incident to a solid (resp. dashed) arc are constrained to have the same (resp. different) color(s), then architectures i.1 are the only ones not 2-colorable.

Observe that the top weight vector of architecture i.1 is constrained to be orthogonal to that of architecture i.0; this is made precise in Prop. 6 in Sec. 5.3. The upshot is that architectures i.0 and i.1 can coincide in function space if and only if their weight matrices vanish– i.e., the architectures degenerate into linear functions. Since neural networks are trained with local optimization, then we think it is unlikely that a $G$-SNN being fit to a nonlinear dataset will degenerate to a linear function at any point in its training; assuming this is true, architectures i.0 and i.1 are effectively confined from one another due to their inequivalent topologies. We discuss this phenomenon in more detail in Sec. 5.3.

## 4.2 The cyclic rotation group

Consider again the group $G = C_6$, but this time a 2D orthogonal representation where each group element acts as a rotation by a multiple of $60°$ on the 2D plane. There are only two irreducible $G$-SNN architectures– one of each type. To visualize these architectures, we set $w = [1, 0]^\top$, $a = 1$, $b = 0.5$, $c = 0$, and $d_* = 0$ in Thm. 4 (b). Based on their contour plots (Fig. 2), we find that the level curves of the type 1 (resp. type 2) architecture are concentric regular dodecagons (resp. hexagons); both architectures are thus clearly invariant to $60°$-rotations.

In the type 2 architecture, the hexagonal level curves increase linearly with radial distance. Since the bias is required to be zero (Eq. 9), a sharp minimum forms at the origin. The architecture has three weight vectors and thus three hidden neurons, and it additionally has a linear term (whose gradient is shown in red in Fig. 2), which—when combined with weight vector 2 using Eq. 2—results in three weight vectors with $C_3$ symmetry. Observe that if we cycle the three hidden neurons of the type 2 architecture , so that each weight vector is rotated three times by $60°$, then we obtain the three weight vectors with reversed orientation; this is a manifestation of the nontrivial topology of the type 2 architecture.

The type 1 architecture has six weight vectors and thus six hidden neurons. Observe that for each weight vector, there is another that is its opposite. Thus, for $[W_* \mid b_*]$ to have pairwise nonparallel rows (see Lemma 2), the bias $b_* = b\vec{1}$ must be nonzero, whence the dodecahedral region in the example $G$-SNN (Fig. 2) where its value plateaus to zero. However, in the asymptotic limit $b \to 0$, the type 1 architecture degenerates to the type 2 architecture but with twice the number of hidden

neurons. Thus, even though the two architectures are topologically distinct, the type 1 architecture can get arbitrarily close to the type 2 architecture in function space (see Supp. C.3 for a richer example—the dihedral rotation group $D_6$—which has irreducible architectures that cannot easily access one another). This has important consequences, which we discuss more in the next section.

## 5 Remarks

### 5.1 Numbers of hidden neurons

The type 1 and type 2 irreducible architectures for the $G = C_6$ rotation group (Fig. 2) have six and three hidden neurons respectively. In addition, the linear term $c_*x$ in the canonical form of a $G$-SNN—if not zero—can be interpreted as two additional hidden neurons. It follows that a general $G$-SNN that is a sum of copies of the two irreducible architectures cannot have $3k + 1$ hidden neurons for any integer $k$. Thus, if we fit a traditional fully-connected SNN with $3k + 1$ hidden neurons to a dataset invariant under $60°$-rotations, then the fit SNN can be a $G$-SNN if and only if one or more of its hidden neurons are redundant– e.g., one hidden neuron is zeroed out, or four hidden neurons sum to form a linear term, leaving $(3k + 1) - 4 = 3(k - 1)$ hidden neurons corresponding to "proper" weight vectors. Although this is a rather simple example, it suggests the possibility of more severe or complicated restrictions on numbers of hidden neurons for larger and richer groups $G$. In these cases, the redundant hidden neurons could perhaps make it more difficult for the SNN to discover the symmetries in the dataset and thus weaken the model, all at the cost of additional computation. We thus conjecture that one factor that determines the optimal number of hidden neurons in traditional SNNs is whether the number admits a $G$-SNN architecture, or—going further—how many different $G$-SNN architectures the number admits.

### 5.2 Network morphisms

Let $f_i^{\mathrm{PZ}} = F^{-1}(\rho_i^{\mathrm{PZ}})$ for $i = 1, 2$ be two $G$-SNN architectures. If for every $f \in f_2^{\mathrm{PZ}}$ there exists a sequence $\{f_n \in f_1^{\mathrm{PZ}}\}_{n=1}^\infty$ that converges to $f$ in the topology of uniform convergence on compact sets,[10] then we say $f_2^{\mathrm{PZ}}$ is *asymptotically included* in $f_1^{\mathrm{PZ}}$ and write $f_2^{\mathrm{PZ}} \hookrightarrow f_1^{\mathrm{PZ}}$. In the cyclic rotation example in Sec. 4.2, as already discussed there, the type 2 architecture is asymptoticly included in the type 1 architecture. In the cyclic permutation example in Sec. 4.1, the asymptotic inclusions[11] furnish a 3-partite topology on the space of irreducible architectures (Fig. 3); here every directed path is an asymptotic inclusion, and the individual arcs could be called "irreducible asymptotic inclusions". This topology provides the necessary structure to perform neural architecture search (NAS), where the irreducible inclusions serve as the *network morphisms* [Wei et al., 2016]. In NAS, network morphisms are used to map underfitting architectures to larger ones, after which training resumes; the upshot is that the larger architecture need not be re-initialized, thereby significantly cutting computation time. In future work, we will run NAS on the space of irreducible $G$-SNN architectures to learn an optimal $G$-SNN in a greedy manner.

Although the above definition of asymptotic inclusion is functional-analytic, the following theorem gives a group-theoretic characterization that is more amenable to computation.

**Theorem 5.** *Let $f_i^{\mathrm{PZ}} = F^{-1}(\rho_{H_i K_i}^{\mathrm{PZ}})$ for $i = 1, 2$ be two irreducible $G$-SNN architectures. Then $f_2^{\mathrm{PZ}} \hookrightarrow f_1^{\mathrm{PZ}}$ iff there exists $(H, K) \in (H_1, K_1)^G$ such that $H \leq H_2$, $K \leq K_2$, and $H \cap K_2 = K$.*

The proof (see Supp. D.1) relies on a non-canonical parameterization of $G$-SNNs; a lemma (Lemma 22) that invokes the Arzelà-Ascoli Theorem; and Thm. (a). Theorem 5 can thus be used to generate network morphisms such as those in Fig. 3 algorithmicly. Observe that as a corollary, since subgroup lattices are connected and subgroup inclusion is transitive, then there are no "isolated" $G$-SNNs; every $G$-SNN architecture is connected to another by some network morphism.

### 5.3 Topological tunneling

---

[10] In this topology, a sequence $\{f_n\}_{n=1}^\infty$ of functions is said to converge to $f$ iff it converges uniformly to $f$ on every compact set in the domain.

[11] These inclusions are "asymptotic" because no two rows of the canonical parameter $[W_* \mid b_*]$ can be exactly antiparallel, preventing the degeneration of one architecture into another.

Recall the discussion in the final paragraph of Sec. 4.1, where we said architectures i.0 and i.1 coincide in function space only when their weight matrices are zero. We see this phenomenon again in the dihedral rotation group example (see Supp. C.3). Both of these are instances of the following proposition, which states that architectures corresponding to distinct cohomology classes are in a sense orthogonal.

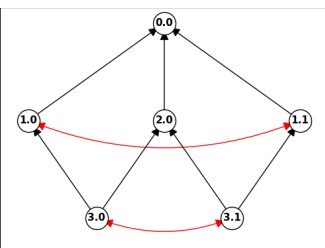

**Proposition 6.** *Let $w_1$ and $w_2$ be the first rows of the canonical weight matrices of two irreducible $G$-SNN architectures $F^{-1}(\rho_{HK_1}^{\mathrm{PZ}})$ and $F^{-1}(\rho_{HK_2}^{\mathrm{PZ}})$ where $K_1 \neq K_2$. Then $w_1^\top w_2 = 0$.*

We call the resulting phenomenon "topological confinement", and it implies that if, for example, Alice generates a nontrivial dataset using architecture 1.1 for $G = C_6$ (Fig. 1) but Bob constructs architecture 1.0 to enforce $G$-invariance

Figure 3: Network morphisms between irreducible $G$-SNN architectures for the cyclic permutation group $G = C_6$. Every directed path in black represents an asymptotic inclusion. Red doubled-arrowed arcs represent the feasibility of topological tunneling. See Supp. C.2 for a richer example– the dihedral permutation group $D_6$.

(as it is one of the more intuitive architectures), then Bob's network will fail to fit to Alice's dataset, even though Bob's network has the right "size"; this suggests that the way we enforce $G$-invariance in a network is important. Rather than randomly selecting between architectures 1.0 and 1.1, or resorting to a larger architecture such as 0.0, we propose to allow "topological tunneling", where the cohomology class of an architecture is transformed by applying the appropriate orthogonal transformation to the top weight vector. This allows us to transform one weight-sharing pattern into another in a way analogous to cutting and regluing a Möbius band to remove the twist. Topological tunneling thus introduces "shortcuts" between certain points in architecture space, hopefully facilitating NAS (Fig. 3). We plan to test this in practice in future work.

# 6 Conclusion

We proved Thm. 4, which gives a classification of all (irreducible) $G$-SNN architectures with ReLU activation for any finite orthogonal group $G$ acting on the input space. The proof is based on a correspondence of every $G$-SNN to a signed perm-rep of $G$ acting on the hidden neurons. We also proved Thm. 5, which characterizes the network morphisms between irreducible $G$-SNN architectures and thus—together with Thm. 4—completely describes $G$-SNN architecture space. A key implication of our theory is the existence of the type 2 $G$-SNN architectures, which to our knowledge have never been explicitly identified in the literature previously.

Various next steps can be taken towards the ultimate goal of $G$-invariant neural architecture design. On one hand, we could try to extend Thm. 4 to deep architectures, which would require us to understand the redundancies of a deep network. We could then investigate the behavior and utility of type 2 symmetry constraints in the context of real deep learning benchmark tasks. on the other hand, we could first go ahead and investigate NAS on $G$-SNNs. For a scalable NAS implementation, we could work out Thm. 4 for specific families of groups to derive more efficient implementations, and we could try to develop an "algebra" of $G$-SNNs where $G$ is a (semi)direct product of smaller groups. Finally, we could consider what are "good" combinations of irreducible $G$-SNN architectures; e.g., which sequences of irreducible architectures converge in sum to universal $G$-invariant approximators fastest? Perhaps answers to these questions could aid in transforming $G$-invariant neural architecture design from an art to a science.

# Acknowledgments and Disclosure of Funding

D.A. and J.O. were supported by DOE grant DE-SC0018175. D.A. was additionally supported by NSF award No. 2202990.

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
