# Supplementary Material

## A  Signed permutation representations

### A.1  Classification up to conjugacy

Let $\{e_1, \ldots, e_n\}$ be the standard orthonormal basis set on $\mathbb{R}^n$. For each $i = 1, \ldots, n$, define $e_{-i} = -e_i$.

For every $B \in \mathcal{PZ}(n)$, let $\psi_B : \mathcal{PZ}(n) \mapsto \mathcal{PZ}(n)$ be the inner automorphism defined by $\psi_B(A) = B^{-1}AB$. Using this notation, two signed perm-reps $\rho, \rho'$ are conjugate if there exists $A \in \mathcal{PZ}(n)$ such that $\rho' = \psi_A \circ \rho$.

The following proposition states that the property of irreducibility is invariant under conjugation, and it thus makes sense to speak of the irreducibility of an entire conjugacy class $\rho^{\mathrm{PZ}}$.

**Proposition 7.** *Let $\rho$ be an irreducible signed perm-rep. Then every signed perm-rep $\rho'$ conjugate to $\rho$ is also irreducible.*

*Proof.* Let $A \in \mathcal{PZ}(n)$ such that $\rho'(g) = A^{-1}\rho(g)A \forall g \in G$. Note that for each $i = 1, \ldots, n$, $Ae_i \in \{e_{\pm 1}, \ldots, e_{\pm n}\}$. Thus, for every $i, j = 1, \ldots, n$, there exists $g \in G$ such that

$$\rho(g)Ae_i = \pm Ae_j$$
$$A^{-1}\rho(g)Ae_i = \pm e_j$$
$$\rho'(g)e_i = \pm e_j.$$

$\square$

We next prove a fundamental lemma that establishes a correspondence between irreducible signed perm-reps and the action of $G$ on certain coset spaces. This is a generalization of the correspondence between ordinary unsigned permutation representations and the action of $G$ on its coset spaces, which is often formalized in terms of the so-called "Burnside ring" [Burnside, 1911, Bouc, 2000]. This lemma is also the basis for the type 1 vs. type 2 dichotomy of irreducible signed perm-reps mentioned in Sec. 2.3.

We require two new definitions first. An *unsigned permutation representation* (unsigned perm-rep) is a signed perm-rep $\rho$ such that $\rho(g) \in \mathcal{P}(n) \forall g \in G$. A signed perm-rep $\rho$ is said to be *transitive* on a set $S \subseteq \mathbb{R}^n$ if for every $v, w \in S$, there exists $g \in G$ such that $\rho(g)v = w$.

**Lemma 8.** *Let $\rho$ be an irreducible signed perm-rep. Define $K \leq H \leq G$ and $U \in \mathcal{Z}(n)$, $U = \mathrm{diag}(u_1, \ldots, u_n)$, by*

$$H = \{g \in G : \rho(g)e_1 = \pm e_1\}$$
$$K = \{g \in G : \rho(g)e_1 = e_1\}$$
$$u_i = \begin{cases} 1, & \text{if } \exists g \in G \mid \rho(g)e_1 = e_i \\ -1, & \text{otherwise.} \end{cases}$$

*Let $\{g_1, \ldots, g_n\}$ be a transversal of $G/H$ such that $U\rho(g_i)Ue_1 = e_i$. For each $i = 1, \ldots, n$, define $g_{-i} = g_i h$ for some $h \in H \setminus K$ if $|H : K| = 2$ and $h = 1$ if $|H : K| = 1$. Then:*

(a) *$|H : K| \leq 2$.*

(b) *If $|H : K| = 1$, then $U\rho(g)Ue_i = e_j$ iff $gg_iK = g_jK$. Moreover, $g \to U\rho(g)U$ is an unsigned perm-rep and is transitive on $\{e_1, \ldots, e_n\}$.*

(c) *If $|H : K| = 2$, then $\rho(g)e_i = e_j$ iff $gg_iK = g_jK$. Moreover, $U = I_n$ and $\rho$ is transitive on $\{\pm e_1, \ldots, \pm e_n\}$.*

*Proof.* **(a)** If there is no $h \in G$ such that $\rho(h)e_1 = -e_1$, then $H = K$, and hence $|H : K| = 1$. On the other hand, suppose there exists $h \in G$ such that $\rho(h)e_1 = -e_1$. Then we have

$$
\begin{aligned}
H &= \{g \in G : \rho(g)e_1 = \pm e_1\} \\
&= \{g \in G : \rho(g)e_1 = e_1\} \cup \{g \in G : \rho(g)e_1 = -e_1\} \\
&= K \cup \{g \in G : \rho(h^{-1})\rho(g)e_1 = -\rho(h^{-1})e_1\} \\
&= K \cup \{g \in G : \rho(h^{-1}g)e_1 = e_1\} \\
&= K \cup hK,
\end{aligned}
$$

and hence $|H : K| = 2$.

**(b)** Suppose $|H : K| = 1$. Let $i, j \in \{\pm 1, \ldots, \pm n\}$, and suppose there exists $g \in G$ such that $U\rho(g)Ue_i = e_j$. We have

$$
\begin{aligned}
U\rho(g)Ue_i &= e_j \\
U\rho(g)UU\rho(g_i)Ue_1 &= U\rho(g_j)Ue_1 \\
\rho(g)\rho(g_i)u_1 e_1 &= \rho(g_j)u_1 e_1 \\
\rho(g_j^{-1}gg_i)e_1 &= e_1 \\
g_j^{-1}gg_i &\in K \\
gg_iK &= g_jK.
\end{aligned}
$$

This sequence of inferences holds in reverse as well, thus establishing the first part of the claim.

Since $|H : K| = 1$, then $g_{-i} = g_i \forall i \in \{1, \ldots, n\}$, and hence the above states that $g \to U\rho(g)U$ is equivalent to the action of $G$ on $\{g_1K, \ldots, g_nK\}$, which is exactly the coset space $G/K$ since $K = H$. By the established equivalence, $g \to U\rho(g)U$ acts transitively on $\{e_1, \ldots, e_n\}$. That $g \to U\rho(g)U$ is an unsigned perm-rep immediately follows from this transitivity.

**(c)** Suppose $|H : K| = 2$. By the same reasoning as in (b), we can establish that $U\rho(g)Ue_i = e_j$ iff $gg_iK = g_jK$. Since $|H : K| = 2$, then clearly $G/K = \{g_{\pm 1}K, \ldots, g_{\pm n}K\}$. We thus have that $g \to U\rho(g)U$ is equivalent to the action of $G$ on $G/K$. By this equivalence, $g \to U\rho(g)U$ acts transitively on $\{\pm e_1, \ldots, \pm e_n\}$. That $U = I_n$ immediately follows from this transitivity, and this in turn implies that $\rho(g)e_i = e_j$ iff $gg_iK = g_jK$ and that $\rho$ acts transitively on $\{\pm e_1, \ldots, \pm e_n\}$. $\square$

**Remark 9.** *In Lemma 8, the irreducibility of the signed perm-rep $\rho$ is necessary to guarantee the existence of $g_i \in G$ such that $U\rho(g_i)Ue_1 = e_i$ for each $i = 1, \ldots, n$.*

**Remark 10.** *In Lemma 8, the signed perm-rep $\rho$ is said to be of* type 1 *(resp.* type 2*) if $|H : K| = 1$ (resp. $|H : K| = 2$).*

We now prove Thm. 1.

*Proof of Thm. 1.* **(a)** Recall by definition of $\mathcal{C}_{\leq 2}^G$, either $|H : K| = 1$ or $|H : K| = 2$. Let $i, j \in \{1, \ldots, n\}$. Since $G$ acts transitively on $G/H$, then there exists $g \in G$ such that $gg_iH = g_jH$. If $|H : K| = 1$, then this is equivalently $gg_iK = g_jK$ so that $\rho_{HK}(g)e_i = e_j$; the rep $\rho_{HK}$ is thus irreducible. If instead $|H : K| = 2$, then we have either $gg_iK = g_jK$ or $gg_iK = g_jhK = g_{-j}K$, so that $\rho_{HK}(g)e_i = e_{\pm j} = \pm e_j$; the rep $\rho_{HK}$ is still irreducible.

**(b)** Let $\rho'$ be an irreducible signed perm-rep. We handle type 1 and type 2 as separate cases.

**(Case 1)** Suppose $\rho'$ is type 1. Then by Lemma 8 (b), $\rho'$ is conjugate to an unsigned perm-rep. We can therefore assume, without loss of generality, that $\rho'$ is an unsigned perm-rep and thus corresponds to the action of $G$ on $G/H'$ for some $H' \leq G$. Let $(H, H)^G \in \mathcal{C}_{\leq 2}^G$ be the unique conjugacy class such that $H$ is conjugate to $H'$. Note that $\rho_{HH}$ is also an unsigned perm-rep and is clearly conjugate to $\rho'$, thus completing the proof for the type 1 case.

**(Case 2)** Suppose $\rho'$ is type 2, and define

$$
\begin{aligned}
H' &= \{g \in G : \rho'(g)e_1 = \pm e_1\} \\
K' &= \{g \in G : \rho'(g)e_1 = e_1\} \\
g_i'K' &= \{g \in G : \rho'(g)e_1 = e_i\} \forall i = 1, \ldots, n.
\end{aligned}
$$

Then there exists a unique $(H, K)^G \in \mathcal{C}^G_{\leq 2}$ and $g_* \in G$ such that

$$H' = g_* H g_*^{-1}$$
$$K' = g_* K g_*^{-1}.$$

Note that since $\rho'$ is type 2, then $|H : K| = |H' : K'| = 2$; thus, $G/K = \{g_{\pm 1}K, \ldots, g_{\pm n}K\}$. Define $\sigma : G \mapsto \{g_{\pm 1}, \ldots, g_{\pm n}\}$ such that $g \in \sigma(g)K$, and define a permutation $\pi$ on $\{\pm 1, \ldots, \pm n\}$ such that

$$g_{\pi(i)} = \sigma(g'_i g_*).$$

Let $A \in \mathcal{PZ}(n)$ such that $Ae_i = e_{\pi(i)}$ for each $i$. Then we claim $\rho' = \psi_A \circ \rho_{HK}$. For any $g \in G$ and $i \in \{\pm 1, \ldots, \pm n\}$, let $j \in \{\pm 1, \ldots, \pm n\}$ such that $\rho'(g)e_i = e_j$. Using Lemma 8 (c), we have

$$\rho'(g)e_i = e_j$$
$$gg'_i K' = g'_j K'$$
$$gg'_i g_* K g_*^{-1} = g'_j g_* K g_*^{-1}$$
$$gg'_i g_* K = g'_j g_* K$$
$$g\sigma(g'_i g_*)K = \sigma(g'_j g_*)K$$
$$gg_{\pi(i)}K = g_{\pi(j)}K$$
$$\rho_{HK}(g)e_{\pi(i)} = e_{\pi(j)}$$
$$\rho_{HK}(g)Ae_i = Ae_j$$
$$A^{-1}\rho_{HK}(g)Ae_i = e_j$$
$$(\psi_A \circ \rho_{HK})(g)e_i = e_j.$$

This sequence of inferences holds in the reverse direction as well, and hence $\rho' = \psi_A \circ \rho_{HK}$ as claimed. □

**Remark 11.** *The transversal $\{g_1, \ldots, g_n\}$ of $G/H$ used in the definition of $\rho_{HK}$ can be recovered from the latter up to $K$. Let $\{g'_1, \ldots, g'_n\}$ be another transversal of $G/H$ such that $\rho_{HK}(g'_i)e_1 = e_i$. By definition of $\rho_{HK}$, $g'_i g_1 K = g_i K$. Since $g_1 \in K$, then $g'_i K = g_i K$.*

## A.2 Some useful properties

For every $z \in \{-1, 1\}^n$, define the signed perm-rep

$$\rho_{HK;z}(g) = \mathrm{diag}(z)\rho_{HK}(g)\,\mathrm{diag}(z) \forall g \in G.$$

The following proposition and subsequent corollary list some useful properties of the $\rho_{HK;z}$. Note that $\rho_{HK} = \rho_{HK;z}$ with $z = \vec{1}$, and hence the statements below hold in particular for the $\rho_{HK}$ as well.

**Proposition 12.** *Let $\rho = \rho_{HK;z}$ be an irreducible signed perm-rep. Let $Z = \mathrm{diag}(z)$. Then the following statements are true:*

*(a) The subgroups $H$ and $K$ satisfy*

$$H = \{g \in G : \rho(g)e_1 = \pm e_1\}$$
$$K = \{g \in G : \rho(g)e_1 = e_1\},$$

*and $\rho$ is of type $|H : K|$.*

*(b) If $\rho$ is type 1 ($|H : K| = 1$), then $\rho_{HK}(g) = Z\rho(g)Z$ is an unsigned perm-rep that acts transitively on $\{e_1, \ldots, e_n\}$.*

*(c) If $\rho$ is type 2 ($|H : K| = 2$), then $\rho$ acts transitively on $\{\pm e_1, \ldots, \pm e_n\}$.*

*Proof.* **(a)** Define the subgroups

$$H' = \{g \in G : \rho(g)e_1 = \pm e_1\}$$
$$K' = \{g \in G : \rho(g)e_1 = e_1\}.$$

We then have

$$
\begin{aligned}
H' &= \{g \in G : Z\rho_{HK}(g)Ze_1 = \pm e_1\} \\
&= \{g \in G : \rho_{HK}(g)Ze_1 = \pm Ze_1\} \\
&= \{g \in G : \rho_{HK}(g)z_1e_1 = \pm z_1 e_1\} \\
&= \{g \in G : \rho_{HK}(g)e_1 = \pm e_1\}.
\end{aligned}
$$

By definition of $\rho_{HK}$ in Thm. 1, we have

$$
\begin{aligned}
H' &= \{g \in G : gg_1K = g_{\pm 1}K\} \\
&= \{g \in G : gK = K \text{ or } gK = hK\} \\
&= K \cup hK \\
&= H.
\end{aligned}
$$

We can similarly show that $K' = K$. By definition of type in Remark 10, $\rho$ is of type $|H' : K'| = |H : K|$.

**(b)** Suppose $\rho$ is type 1 so that $|H : K| = 1$ and hence $H = K$. Then $\rho_{HK}(g)e_i = e_j$ iff $gg_iK = g_jK$, where $\{g_1, \ldots, g_n\}$ is the transversal of $G/H$ used in the definition of $\rho_{HK}$ in Thm. 1. Since $H = K$, however, $\{g_1, \ldots, g_n\}$ is equivalently a transversal of $G/K$, and hence we see that the action of $\rho_{HK}$ is equivalent to the action of $G$ on $G/K$. As in the proof of Lemma 8 (b), this implies the claim.

**(c)** Suppose $\rho$ is type 2. Then the claim immediately follows by Lemma 8 (c). □

The following corollary results from the combination of Lemma 8 and Prop. 12.

**Corollary 13.** *Let $\rho$ be an irreducible signed perm-rep. Define $K \le H \le G$ and $z \in \{-1, 1\}^n$ by*

$$
\begin{aligned}
H &= \{g \in G : \rho(g)e_1 = \pm e_1\} \\
K &= \{g \in G : \rho(g)e_1 = e_1\} \\
z_i &= \begin{cases} 1, & \text{if } \exists g \in G \mid \rho(g)e_1 = e_i \\ -1, & \text{otherwise.} \end{cases}
\end{aligned}
$$

*Then $\rho = \rho_{HK;z}$. Moreover, if $\rho$ is type 2 ($|H : K| = 2$), then $z = \vec{1}$ so that $\rho = \rho_{HK}$.*

*Proof.* If $\rho$ is type 2, then by Lemma 8 (c), $\rho$ is transitive on $\{\pm e_1, \ldots, \pm e_n\}$ so that $z_i = 1$ for each $i = 1, \ldots, n$. Now let $\rho_z(g) = \operatorname{diag}(z)\rho(g)\operatorname{diag}(z) \forall g \in G$. Then again by Lemma 8, $\rho_z(g)e_i = e_j$ iff $gg_iK = g_jK$; however, recalling Thm. 1, this is identical to the definition of $\rho_{HK}$. Hence, $\rho_z = \rho_{HK}$, or equivalently $\rho = \rho_{HK;z}$. □

### A.3 Group cohomology

For every signed perm-rep $\rho_{HK}$, let $\pi_H : G \mapsto \mathcal{P}(n)$ and $\omega_{HK} : G \mapsto \mathcal{Z}(n)$ be the unique functions satisfying $\rho_{HK}(g) = \omega_{HK}(g)\pi_H(g) \forall g \in G$.[12] The following proposition justifies the notation $\pi_H$; i.e., $\pi_H$ does not depend on the choice of $K$ and $z$.

**Proposition 14.** *Let $\rho_{HK}$ be an irreducible signed perm-rep of $G$, and let $\pi_{HK} : G \mapsto \mathcal{P}(n)$ and $\zeta_{HK} : G \mapsto \mathcal{Z}(n)$ be the unique functions satisfying $\rho_{HK}(g) = \pi_{HK}(g)\zeta_{HK}(g) \forall g \in G$. Then $\pi_{HK}$ is independent of $K$.*

*Proof.* As in Remark 11, let $\{g_1, \ldots, g_n\}$ be a transversal of $G/H$ such that $\rho_{HK}(g_i)e_1 = e_i$ for $i = 1, \ldots, n$. In general, $g_1 \in K$; however, without loss of generality, assume $g_1 = 1$ so that it is independent of $K$. If $g \in G$ and $i, j \in \{\pm 1, \ldots, \pm n\}$ such that $\rho_{HK}(g)e_i = e_j$, then define $\pi_{HK}$ and $\zeta_{HK}$ such that

$$
\begin{aligned}
\pi_{HK}(g)e_i &= e_{|j|} \\
\zeta_{HK}(g)e_i &= \operatorname{sign}(j).
\end{aligned}
$$

---

[12] By uniqueness of factorization in a semidirect product, there exist unique functions $\pi : G \mapsto \mathcal{P}(n)$ and $\zeta, \omega : G \mapsto \mathcal{Z}(n)$ such that $\rho(g) = \pi(g)\zeta(g) = \omega(g)\pi(g) \forall g \in G$.

It is then easy to verify that $\rho_{HK}(g) = \pi_{HK}(g)\zeta_{HK}(g)\forall g \in G$; hence by uniqueness, these are the correct definitions of $\pi_{HK}$ and $\zeta_{HK}$. By these definitions, for $g \in G$ and $i, j \in \{1, \ldots, n\}$, $\pi_{HK}(g)e_i = e_j$ iff $gg_iK = g_{\pm j}K$, which in turn holds iff $gg_iH = g_jH$. This reveals that $\pi_{HK}$ does not depend on $K$ but only $H$. $\qquad\square$

The following proposition relates the structure of irreducible signed perm-reps of $G$ to its cohomology.

**Proposition 15.** *For every conjugacy class $\rho_{HK}^{\mathrm{PZ}}$ of irreducible signed perm-reps, define the $G$-module $M_H = (\{0, 1\}^n, \pi_H)$ under addition modulo 2, where $n = |G|/|H|$. Define $\hat{\omega}_{HK} : G \mapsto M_H$ such that $\hat{\omega}_{HK}(g) = \frac{1}{2}[I - \mathrm{diag}(\omega_{HK}(g))]$. Then:*

    *(a) The first cohomology group of $G$ with coefficients in $M_H$ is given by[13]*

$$\mathcal{H}^1(G, M_H) = \{[hat\omega_{HK}] : K \le H \mid |H : K| \le 2\}_{\ne},$$

    *where $[\hat{\omega}_{HK}]$ is the set of all cocycles cohomologous to $\omega_{HK}$, and where the addition operation satisfies*

$$[\hat{\omega}_{HK}] = [\hat{\omega}_{HK_1}] + [\hat{\omega}_{HK_2}] \Leftrightarrow K = K_1 \cap K_2 \cup ((H \setminus K_1) \cap (H \setminus K_2)).$$

    *(b) The partition of the first cohomology group into orbits under the action of the $G$-module automorphism group $\mathrm{aut}(M_H)$ is given by*

$$H^1(G, M_H)/\mathrm{aut}(M_H) = \{\{[\hat{\omega}_{HK'}] : (H, K') \in (H, K)^G\} : (H, K)^G \in \mathcal{C}_{\le 2}^G\}.$$

    *(c) $\rho_{HK}$ is type 1 if and only if $\hat{\omega}_{HK}$ is in the zero cohomology class.*

*Proof of Prop. 15.* **(a)** We first show that every $\hat{\omega}_{HK}$ is a 1-cocycle by verifying the cocycle condition. For $g_1, g_2 \in G$, we have

$$
\begin{aligned}
\omega_{HK}(g_1g_2)\pi_H(g_1g_2) &= \rho_{HK}(g_1g_2) \\
&= \rho_{HK}(g_1)\rho_{HK}(g_2) \\
&= \omega_{HK}(g_1)\pi_H(g_1)\omega_{HK}(g_2)\pi_H(g_2) \\
&= \omega_{HK}(g_1)\pi_H(g_1)\omega_{HK}(g_2)\pi_H(g_1)^\top\pi_H(g_1)\pi_H(g_2).
\end{aligned}
$$

Equating the factors contained in $\mathcal{Z}(n)$, we have

$$\omega_{HK}(g_1g_2) = \omega_{HK}(g_1)\pi_H(g_1)\omega_{HK}(g_2)\pi_H(g_1)^\top.$$

Writing this in terms of vectors in $M_H$, we obtain the 1-cocycle condition:

$$\hat{\omega}_{HK}(g_1g_2) = \hat{\omega}_{HK}(g_1) + \pi_H(g_1)\hat{\omega}_{HK}(g_2),$$

and hence $\hat{\omega}_{HK}$ is a 1-cocycle.

Next, before proving the main claim, we characterize all cocycles cohomologous to $\hat{\omega}_{HK}$. For every $z \in \{-1, 1\}^n$, define the signed perm-rep $\rho_{HK;z}(g) = \mathrm{diag}(z)\rho_{HK}(g)\mathrm{diag}(z)\forall g \in G$, and let $\pi_{H;z} : G \mapsto \mathcal{P}(n)$ and $\omega_{HK;z} : G \mapsto \mathcal{Z}(n)$ be the unique functions satisfying $\rho_{HK;z}(g) = \omega_{HK;z}(g)\pi_{H;z}(g)\forall g \in G$. We have for all $g \in G$,

$$
\begin{aligned}
\rho_{HK;z}(g) &= \mathrm{diag}(z)\rho_{HK}(g)\mathrm{diag}(z) \\
\omega_{HK;z}(g)\pi_{H;z}(g) &= \mathrm{diag}(z)\omega_{HK}(g)\pi_H(g)\mathrm{diag}(z) \\
&= \mathrm{diag}(z)\omega_{HK}(g)\pi_H(g)\mathrm{diag}(z)\pi_H(g)^\top\pi_H(g).
\end{aligned}
$$

Equating factors in $\mathcal{P}(n)$ and equating factors in $\mathcal{Z}(n)$, we obtain

$$\pi_{H;z}(g) = \pi_H(g)$$

$$\omega_{HK;z}(g) = \mathrm{diag}(z)\omega_{HK}(g)\pi_H(g)\mathrm{diag}(z)\pi_H(g)^\top.$$

---

[13]We use the notation $\{\ldots\}_{\ne}$ to emphasize that, during the construction of the set, the enumerated elements are distinct.

The first of these equations tells us that $\pi_{H;z}$ is independent of $z$, and we will thus omit the subscript $z$ in $\pi_{H;z}$ henceforth. Writing the second of these equations in terms of vectors in $M_H$, we have

$$\hat{\omega}_{HK;z}(g) = \hat{z} + \hat{\omega}_{HK}(g) + \pi_H(g)\hat{z}$$
$$= (\pi_H(g)\hat{z} - \hat{z}) + \hat{\omega}_{HK}(g),$$

where we used the fact that $\hat{z} = -\hat{z} \pmod 2$. Since $g \to \pi(g)\hat{z} - \hat{z}$ is a coboundary, then $\hat{\omega}_{HK;z}$ is cohomologous to $\hat{\omega}_{HK}$; from the above, the converse is also easily verified.

We thus have

$$[\hat{\omega}_{HK}] = \{\hat{\omega}_{HK;z} : z \in \{-1, 1\}^n\},$$

where distinct $z$ do not necessarily imply distinct $\hat{\omega}_{HK;z}$.

We now prove the main claim. We first prove that the cohomology classes $[\hat{\omega}_{HK}]$ enumerated over all $K \le H \mid |H : K| \le 2$ are distinct. Suppose $[\hat{\omega}_{HK_1}] = [\hat{\omega}_{HK_2}]$; i.e., $\hat{\omega}_{HK_1}$ and $\hat{\omega}_{HK_2}$ are cohomologous. We will show $K_1 = K_2$. By the above, there exists $z \in \{-1, 1\}^n$ such that $\hat{\omega}_{HK_2} = \hat{\omega}_{HK_1;z}$; Converting this back in terms of diagonal matrices and multiplying the resulting equation from the right by $\pi_H$, we obtain $\rho_{HK_2} = \rho_{HK_1;z}$. By definition of $\rho_{HK_2}$, we have

$$\{g \in G : \rho_{HK_2}(g)e_1 = e_1\} = K_2.$$

On the other hand,

$$\begin{aligned}
\{g \in G : \rho_{HK_2}(g)e_1 = e_1\} &= \{g \in G : \rho_{HK_1;z}(g)e_1 = e_1\} \\
&= \{g \in G : \operatorname{diag}(z)\rho_{HK_1}(g)\operatorname{diag}(z)e_1 = e_1\} \\
&= \{g \in G : \rho_{HK_1}(g)\operatorname{diag}(z)e_1 = \operatorname{diag}(z)e_1\} \\
&= \{g \in G : \rho_{HK_1}(g)z_1e_1 = z_1e_1\} \\
&= \{g \in G : \rho_{HK_1}(g)e_1 = e_1\} \\
&= K_1.
\end{aligned}$$

Ergo, $K_1 = K_2$.

We next prove that every 1-cocycle is contained in one of the cohomology classes $[\hat{\omega}_{HK}]$. Let $\hat{\omega} : G \mapsto M_H$ be a 1-cocycle. Then $\rho(g) = \omega(g)\pi_H(g)\forall g \in G$ defines an irreducible signed perm rep. It is easy to verify that

$$H = \{g \in G : \rho(g)e_1 = \pm e_1\},$$

and define

$$K = \{g \in G : \rho(g)e_1 = e_1\}.$$

Then by Cor. 13, $\rho = \rho_{HK;z}$ for some $z \in \{-1, 1\}^n$, and hence $\hat{\omega} = \hat{\omega}_{HK;z}$ so that $\hat{\omega} \in [\hat{\omega}_{HK}$.

All that is left for (a) is to prove the claimed identity for the addition operation. First, however, given a cocycle $\hat{\omega}_{HK;z}$, note that by Prop. 12 (a), we have

$$\begin{aligned}
K &= \{g \in G : \rho_{HK;z}(g)e_1 = e_1\} \\
&= \{g \in G : \omega_{HK;z}(g)\pi_H(g)e_1 = e_1\} \\
&= \{g \in H : \omega_{HK;z}(g)e_1 = e_1\} \\
&= \{g \in H : \omega_{HK;z}(g)_{11} = 1\} \\
&= \{g \in H : \hat{\omega}_{HK;z}(g)_1 = 0\}.
\end{aligned}$$

Now consider the sum of two cohomology classes $[\hat{\omega}_{HK_1}]$ and $[\hat{\omega}_{HK_2}]$. Since we have established all elements of the cohomology group, then we know that there exists $K \le H \mid |H : K| \le 2$ such that

$$[\hat{\omega}_{HK}] = [\hat{\omega}_{HK_1}] + [\hat{\omega}_{HK_2}].$$

Thus, there exists $z \in \{-1, 1\}^n$ such that

$$\hat{\omega}_{HK;z} = \hat{\omega}_{HK_1} + \hat{\omega}_{HK_2}.$$

Now by the above, we have

$$\begin{aligned}
K &= \{g \in H : \hat{\omega}_{HK;z}(g)_1 = 0\} \\
&= \{g \in H : \hat{\omega}_{HK_1}(g)_1 + \hat{\omega}_{HK_2}(g)_1 = 0\} \\
&= \{g \in H : \hat{\omega}_{HK_1}(g)_1 = \hat{\omega}_{HK_2}(g)_1 = 0\} \cup \{g \in H : \hat{\omega}_{HK_1}(g)_1 = \hat{\omega}_{HK_2}(g)_1 = 1\} \\
&= K_1 \cap K_2 \cup ((H \setminus K_1) \cap (H \setminus K_2)),
\end{aligned}$$

thereby establishing the claim.

**(b)** Let $[\hat{\omega}_{HK_1}]$ and $[\hat{\omega}_{HK_2}]$ be two cohomology classes. We must show $(H, K_1)$ is conjugate to $(H, K_2)$ if and only if there exists $P \in \mathcal{P}(n)$ such that $[P, \pi(g)] = 0 \forall g \in G$ and $[P\hat{\omega}_{HK_1}] = [\hat{\omega}_{HK_2}]$. Suppose $(H, K_1)$ and $(H, K_2)$ are conjugate. Then by Thm. 1, $\rho_{HK_1}$ and $\rho_{HK_2}$ are conjugate, so that there exist $P \in \mathcal{P}(n)$ and $Z \in \mathcal{Z}(n)$, $Z = \mathrm{diag}(z)$, such that for all $g \in G$,

$$\rho_{HK_2}(g) = ZP\rho_{HK_1}(g)(ZP)^{-1}$$
$$\rho_{HK_2}(g) = ZP\rho_{HK_1}(g)P^\top Z$$
$$\rho_{HK_2;z}(g) = P\rho_{HK_1}(g)P^\top$$
$$\omega_{HK_2;z}(g)\pi_H(g) = P\omega_{HK_1}(g)\pi_H(g)P^\top$$
$$\omega_{HK_2;z}(g)\pi_H(g) = P\omega_{HK_1}(g)P^\top P\pi_H(g)P^\top.$$

Equating the factors in $\mathcal{P}(n)$ and the factors in $\mathcal{Z}(n)$, we obtain

$$\pi_H(g) = P\pi_H(g)P^\top$$
$$\omega_{HK_2;z}(g) = P\omega_{HK_1}(g)P^\top.$$

The first of these equations establishes the commutation $[P, \pi(g)] = 0$. The second equation implies

$$\hat{\omega}_{HK_2;z}(g) = P\hat{\omega}_{HK_1}(g)$$
$$[\hat{\omega}_{HK_2}] = [P\hat{\omega}_{HK_1}].$$

The above steps can be reversed to prove the converse.

**(c)** For every $K \leq H \mid |H : K| \leq 2$, observe that

$$K \cap H \cup ((H \setminus K) \cap (H \setminus H)) = K.$$

By (a), $[\hat{\omega}_{HK}]$, $H = K$, is thus the zero cohomology class. Therefore, $\rho_{HK;z}$ is type 1 ($|H : K| = 1$, or $H = K$) if and only if $[\hat{\omega}_{HK}]$ is the zero cohomology class. $\qquad\square$

The type 1 vs. type 2 dichotomy is thus rooted in whether a signed perm-rep "twists" over $G/H$. Proposition 15 also lets us interpret the notation $\rho_{HK}$: The subgroup $H$ determines the coefficient module $M_H$ and hence the cohomology ring, and the subgroup $K$ determines the cohomology class in $\mathcal{H}^1(G, M_H)$.

# B  Classification of $G$-SNNs

## B.1  Canonical parameterization

Let $f : \mathbb{R}^m \mapsto \mathbb{R}$ be a continuous piecewise-affine function. An *affine region* $X \subseteq \mathbb{R}^m$ of $f$ is a maximal polytope over which $f$ is affine.

Let $f : \mathbb{R}^m \mapsto \mathbb{R}$ be an SNN of the form in Eq. 1, and note that $f$ is a continuous piecewise-affine function. Then the *signature* $r$ of an affine region $X \subseteq \mathbb{R}^m$ is the binary vector $r = H(Wx + b)$, for any arbitrary choice of $x$ in the interior of $X$ and where $H$ is the Heaviside step function (where we set $H(0) = 0$).

The following small proposition establishes the identity given in Eq. 2.

**Proposition 16.** *For all $x \in \mathbb{R}$ and $z \in \{-1, 1\}$,*

$$\mathrm{ReLU}(x) - \mathrm{ReLU}(zx) = H(-z)x.$$

*Proof.* It is easy to verify that $\mathrm{ReLU}(x) - \mathrm{ReLU}(-x) = x$ for all $x$. Now we have two cases:

**Case 1 ($z = -1$)** We have

$$\begin{aligned}
\mathrm{ReLU}(x) - \mathrm{ReLU}(zx) &= \mathrm{ReLU}(x) - \mathrm{ReLU}(-x) \\
&= x \\
&= H[-(-1)]x \\
&= H(-z)x.
\end{aligned}$$

**Case 2** ($z = 1$) We have

$$
\mathrm{ReLU}(x) - \mathrm{ReLU}(zx)
$$
$$
\mathrm{ReLU}(x) - \mathrm{ReLU}(x)
$$
$$
= 0
$$
$$
= H(-1)x
$$
$$
= H(-z)x.
$$

$\square$

We now prove Lemma 2.

*Proof of Lemma 2.* Given access to the data $D = \{(x, f(x)) : x \in \mathbb{R}^m\}$, we will show that we can in principle determine $[W_* \mid b_*], a_*, c_*, d_*$ uniquely. Since $f$ admits the form in Eq. 1, the set of points at which $f$ is not differentiable is a union of $n_*$ distinct affine spaces each of dimension $m - 1$, for a unique $n_* \leq n$. From the data $D$, we can in principle determine the equation of each affine space; let $w_{*i}^\top x + b_{*i} = 0$ be the equation defining the $i$th affine space, where $\|w_{*i}\| = 1$. Let $W_* \in \mathbb{R}^{n_* \times m}$ with $i$th row $w_{*i}^\top$ and $b_* \in \mathbb{R}^{n_*}$ with elements $b_{*i}$. Note that no two rows of $[W_* \mid b_*]$ are parallel, as parallel rows would correspond to the same affine space. Thus, $[W_* \mid b_*] \in \Theta_{n_*}$. Note that the action of any element in $\mathrm{PZ}(n_*)$ on $[W_* \mid b_*]$ leaves the corresponding set of affine spaces invariant; we thus assume, without loss of generality, that $[W_* \mid b_*] \in \Omega_{n_*}$, thereby establishing the uniqueness of $[W_* \mid b_*]$. The function $f$ now admits the form

$$
f(x) = a_*^\top \mathrm{ReLU}[Z(W_* x + b_*)] + \tilde{f}(x),
$$

for some $a_* \in \mathbb{R}^{n_*}$, $Z \in \mathcal{Z}(n_*)$, and some differentiable piecewise affine function $\tilde{f}(x) : \mathbb{R}^m \mapsto \mathbb{R}$. Note that $a_{*i} \neq 0$ for each $i = 1, \ldots, n_*$; otherwise, we could simply delete the $i$th row of $[W_* \mid b_*]$. Since $\tilde{f}$ is both piecewise-affine and differentiable, then it is necessarily affine; hence there exist $\tilde{c}_* \in \mathbb{R}^m$ and $\tilde{d}_* \in \mathbb{R}$ such that $\tilde{f}(x) = \tilde{c}_*^\top x + \tilde{d}_*$ and thus

$$
f(x) = a_*^\top \mathrm{ReLU}[Z(W_* x + b_*)] + \tilde{c}_*^\top x + \tilde{d}_*.
$$

Now applying Prop. 16, we have

$$
f(x) = a_*^\top \mathrm{ReLU}(W_* x + b_*) - a_*^\top H(-Z)(W_* x + b_*) + \tilde{c}_* x + \tilde{d}_*
$$
$$
= a_*^\top \mathrm{ReLU}(W_* x + b_*) + c_* x + d_*,
$$

where we define

$$
c_* = \tilde{c}_* - a_*^\top H(-Z) W_*
$$
$$
d_* = \tilde{d}_* - a_*^\top H(-Z) b_*.
$$

All that remains is to show $a_*$, $c_*$, and $d_*$ are unique. We start by showing $a_*$ is unique. Consider two adjacent affine regions $X, X' \subset \mathbb{R}^m$ of $f$, where the shared boundary is defined by the $i$th affine space. Let $r$ and $r'$ be the signatures of $X$ and $X'$. Letting $x$ and $x'$ be two arbitrary points from the interiors of $X$ and $X'$ respectively, we have the following difference of gradients with respect to $x$:

$$
\nabla f(x') - \nabla f(x) = W_*^\top \mathrm{diag}(r') a_* - W_*^\top \mathrm{diag}(r) a_*
$$
$$
= W_*^\top \mathrm{diag}(r' - r) a_*.
$$

Since $X$ and $X'$ differ only across the $i$th affine space, then all entries of $r' - r$ are zero except the $i$th entry. We therefore have

$$
\nabla f(x') - \nabla f(x) = a_{*i}(r_i' - r_i) w_{*i}.
$$

Since $r' - r$ and $w_{*i}$ are nonzero and unique, then we can in principle solve this equation to determine a unique value for $a_{*i}$.

To show $c_*$ is unique, we recall the gradient of $f$ evaluated at the point of differentiability $x \in X$:

$$
\nabla f(x) = W_*^\top \mathrm{diag}(r) a_* + c_*.
$$

Since $a_*$ and $W_*$ have been determined, then we can in principle solve this equation to determine a unique value for $c_*$. Once this is done, we can then evaluate $f$ at $x$ and solve for the only remaining unknown $d_*$, thereby determining a unique value for $d_*$ as well. $\square$

**Remark 17.** *Suppose $f : \mathbb{R}^m \mapsto \mathbb{R}$ admits the form in Eq. 1. Then it is possible for there to exist $i$ and $j$ such that $a_i = -a_j$, $w_i = -w_j$, and $b_i = -b_j$. In this case, we have*

$$a_i \operatorname{ReLU}(w_i^\top x + b_i) + a_j \operatorname{ReLU}(w_j^\top x + b_j) = a_i \operatorname{ReLU}(w_i^\top x + b_i) - a_i \operatorname{ReLU}[-(w_i^\top x + b_i)]$$
$$= a_i(w_i^\top x + b_i),$$

*which follows from Prop. 16. Such affine and differentiable terms can thus arise, which is why we include the $c_* x + d_*$ term in Lemma 2. Moreover, observe that because $[W_* \mid b_*] \in \Theta_{n_*}$ in Lemma 2, no two rows of $[W_* \mid b_*]$ are equal or opposites of one another, and thus no two hidden neurons can be combined to yield an affine term; all affine terms are thus collected in the $c_* x + d_*$ term, which helps to make the canonical form of $f$ unique.*

In general, given a group action on a set, the existence of a fundamental domain is not guaranteed. The next proposition guarantees the existence of a fundamental domain in $\Theta_n$ under the action of $\mathrm{PZ}(n)$ by way of a constructive example.

**Proposition 18.** *Let $\leq_*$ be a total order on $\mathbb{R}^{m+1}$. Let $\Omega$ be the set of all $[W \mid b] \in \Theta_n$ such that the first nonzero entry of each row of $W$ is positive and the rows of $[W \mid b]$ are sorted in ascending order under $\leq_*$. Then $\Omega$ is a fundamental domain.*

*Proof.* We will show that $\{A\Omega : A \in \mathrm{PZ}(n)\}$ is a partition of $\Theta_n$. First, however, let $[W \mid b] \in \Omega$. Since the rows of $[W \mid b]$ are nonzero (since the rows of $W$ have unit norm), then the action of any non-identity $Z \in \mathcal{Z}(n)$ sends $[W \mid b]$ out of $\Omega$. Similarly, since the rows of $[W \mid b]$ are pairwise nonparallel and in particular distinct, then any non-identity $P \in \mathcal{P}(n)$ breaks the ascending order of the rows of $[W \mid b]$ and sends it out of $\Omega$. Finally, since no two rows of $[W \mid b]$ are opposites, then the actions of $P$ and $Z$ cannot cancel one another. It thus follows that every $A \in \mathrm{PZ}(n)$ sends $[W \mid b]$ out of $\Omega$.

We now proceed to show the elements in the claimed partition are disjoint. Let $A, B \in \mathrm{PZ}(n)$, and suppose $A\Omega \cap B\Omega \neq \emptyset$. So, let $[W \mid b] \in A\Omega \cap B\Omega$. Thus, $A^{-1}[W \mid b]$ and $B^{-1}[W \mid b]$ are both in $\Omega$. We also note $(B^{-1}A)A^{-1}[W \mid b] = B^{-1}[W \mid b]$. If $B^{-1}A$ is not the identity, then by the above, it sends $A^{-1}[W \mid b]$ out of $\Omega$, so that $B^{-1}[W \mid b] \notin \Omega$. Since, however, $B^{-1}[W \mid b] \in \Omega$, then $B^{-1}A = I$ so that $A = B$.

We next show that every $[W \mid b] \in \Theta_n$ belongs to some element of the claimed partition. Clearly, there exists $A \in \mathrm{PZ}(n)$ such that $A[W \mid b] \in \Omega$, so that $[W \mid b] \in A^{-1}\Omega$. $\square$

### B.2 $G$-SNNs and signed perm-reps

We prove Lemma 3.

*Proof of Lemma 3.* We only prove the forward implication; the converse is then straightforward to verify. We write $f$ in its canonical form:

$$f(x) = a_*^\top \operatorname{ReLU}(W_* x + b_*) + c_*^\top x + d_*.$$

Let $g \in G$. Since $g$ is orthogonal and each row of $W_*$ has unit norm, then so does each row of $W_* g$. Moreover, since the transformation $[W_* \mid b_*] \to [W_* g \mid b_*]$ is invertible and no two rows of $[W_* \mid b_*]$ are parallel, then the same is true for the rows of $[W_* g \mid b_*]$. Thus, $[W_* g \mid b_*] \in \Theta_{n_*}$, and hence there exists a unique matrix $\rho(g) \in \mathrm{PZ}(n_*)$ such that $[W_* g \mid b_*] \in \rho(g)\Omega_{n_*}$. Let $\pi(g) \in \mathcal{P}(n_*)$ and $\zeta(g) \in \mathcal{Z}(n_*)$ such that $\rho(g) = \pi(g)\zeta(g)$. We have

$$\begin{aligned}
f(gx) &= a_*^\top \operatorname{ReLU}(W_* gx + b_*) + c_*^\top gx + d_* \\
&= a_*^\top \operatorname{ReLU}[\rho(g)(\rho(g)^{-1}W_* gx + \rho(g)^{-1}b_*)] + c_*^\top gx + d_* \\
&= a_*^\top \operatorname{ReLU}[\pi(g)\zeta(g)(\rho(g)^{-1}W_* gx + \rho(g)^{-1}b_*)] + c_*^\top gx + d_* \\
&= a_*^\top \pi(g) \operatorname{ReLU}[\zeta(g)(\rho(g)^{-1}W_* gx + \rho(g)^{-1}b_*)] + c_*^\top gx + d_*.
\end{aligned}$$

Using Prop. 16, this is

$$\begin{aligned}
f(gx) &= a_*^\top \pi(g) \operatorname{ReLU}(\rho(g)^{-1}W_* gx + \rho(g)^{-1}b_*) - a_*^\top H(-\zeta(g))(\rho(g)^{-1}W_* x + \rho(g)^{-1}b_*) + c_*^\top gx + d_* \\
&= a_*^\top \pi(g) \operatorname{ReLU}(\rho(g)^{-1}W_* gx + \rho(g)^{-1}b_*) + [c_*^\top g - a_*^\top H(-\zeta(g))\rho(g)^{-1}W_*]x + [d_* - a_*^\top H(-\zeta(g))\rho(g)^{-1}b_*].
\end{aligned}$$

Note that $\rho(g)^{-1}[W_* \mid b_*] \in \Omega_{n_*}$. Since $f$ is $G$-invariant, then $f(gx) = f(x) \forall x \in \mathbb{R}^m$. By uniqueness of canonical parameters with respect to the fundamental domain $\Omega_{n_*}$ (Lemma 2), the canonical parameters of the SNNs $f$ and $f \circ g$ must be equal. We thus obtain the constraints

$$W_* = \rho(g)^{-1} W_* g$$
$$a_*^\top = a_*^\top \pi(g)$$
$$b_* = \rho(g)^{-1} b_*$$
$$c_*^\top = c_*^\top g - a_*^\top H(-\zeta(g))\rho(g)^{-1} W_* g$$
$$d_* = d_* - a_*^\top H(-\zeta(g))\rho(g)^{-1} b_*.$$

The first three constraints are clearly equivalent to the ones on $W_*$, $a_*$, and $b_*$ claimed in the lemma statement; we thus take these as established. By the established $W_*$ and $b_*$ constraints, the $c_*$ and $d_*$ constraints simplify to

$$c_*^\top = c_*^\top g - a_*^\top H(-\zeta(g)) W_*$$
$$d_* = d_* - a_*^\top H(-\zeta(g)) b_*.$$

Now since $\zeta(g)$ is a diagonal matrix with $\pm 1$ along its diagonal, then we have

$$H(-\zeta(g)) = \frac{1}{2}(I - \zeta(g)).$$

Using the established $a_*$ constraint, we have

$$
\begin{aligned}
a_*^\top H(-\zeta(g)) &= \frac{1}{2} a_*^\top (I - \zeta(g)) \\
&= \frac{1}{2} a_*^\top (I - \pi(g)\zeta(g)) \\
&= \frac{1}{2} a_*^\top (I - \rho(g)).
\end{aligned}
$$

By the established $b_*$ constraint, we have $(I - \rho(g))b_* = b_* - b_* = 0$; we thus see that the above constraint on $d_*$ is trivially satisfied. By the established $W_*$ constraint, the constraint on $c_*$ becomes

$$
\begin{aligned}
c_*^\top &= c_*^\top g - \frac{1}{2} a_*^\top (I - \rho(g)) W_* \\
&= c_*^\top g - \frac{1}{2} a_*^\top W_* (I - g) \\
g^\top c_* &= c_* + \frac{1}{2}(I - g^\top) W_*^\top a_*.
\end{aligned}
$$

Since this holds for all $g \in G$, then we may substitute $g^\top$ with $g$ to establish the claimed constraint on $c_*$.

Finally, we prove that $\rho : G \mapsto \mathrm{PZ}(n_*)$ is a homomorphism. Let $g_1, g_2 \in G$. By the established constraint on $W_*$, we have

$$
\begin{aligned}
\rho(g_1)\rho(g_2)[W_* \mid b_*] &= \rho(g_1)[W_* g_2 \mid b_*] \\
&= [W_* g_1 g_2 \mid b_*].
\end{aligned}
$$

On the other hand, by definition of $\rho$, we have $\rho(g_1 g_2)[W_* \mid b_*] = [W_* g_1 g_2 \mid b_*]$. Thus, $[W_* g_1 g_2 \mid b_*]$ is thus an element of both $\rho(g_1 g_2)^{-1}\Omega_{n*}$ and of $[\rho(g_1)\rho(g_2)]^{-1}\Omega_{n*}$, which in turn implies $\rho(g_1 g_2) = \rho(g_1)\rho(g_2)$. □

The next proposition states that every $G$-SNN can be written as a sum of irreducible $G$-SNNs, thereby simplifying the classification problem of $G$-SNNs to that only of irreducible $G$-SNNs. Recall the notation introduced in Sec. 3.2.

**Proposition 19.** *Every $G$-SNN admits a decomposition into a sum of irreducible $G$-SNNs.*

*Proof.* Let $f \in \mathrm{SNN}(G)$, and let $\rho \in F(f)$. Let $n_*$ be the degree of $\rho$; i.e., the number of rows of the canonical weight matrix of $f$. Then partition $\{e_1, \ldots, e_{n*}\}$ into orbits such that $e_i$ and $e_j$ belong

to the same orbit if and only if there exists $g \in G$ such that $\rho(g)e_i = \pm e_j$. Without loss of generality, select $\rho \in F(f)$ such that each orbit consists of consecutive elements (this is done by an appropriate conjugation of $\rho$); i.e., each orbit has the form $\{e_i, e_{i+1}, \dots, e_{i+j}\}$. Now write $f$ in canonical form such that the corresponding signed perm-rep by Lemma 3 is $\rho$:

$$f(x) = a_*^\top \operatorname{ReLU}(W_* x + b_*) + c_*^\top x + d_*.$$

For each $i = 1, \dots, k$, where we have $k$ orbits, define the $G$-SNN $f_i$ by taking only the elements $a_{*j}$ of $a_*$, rows $w_{*j}^\top$ of $W_*$, and elements $b_{*j}$ of $b_*$ such that $e_j$ belongs to the $i$th orbit; include the affine term $c_*^\top x + d_*$ only in $f_k$. Then clearly $f = f_1 + \dots f_k$, where each $f_i$ is $G$-invariant and irreducible. $\qquad\square$

## B.3 The classification theorem

This section gives a proof for Thm. 4. Recall the following notation introduced in Sec. 3.3: If $A$ is a linear operator (resp. set of linear operators), then let $P_A$ be the orthogonal projection operator onto the vector subspace that is pointwise-invariant under the action of $A$ (resp. all elements of $A$). Note that if $A$ is a finite orthogonal group, then [Serre, 1977, sec. 2.6]

$$P_A = \frac{1}{|A|} \sum_{a \in A} a.$$

In addition, if $P_1, P_2$ are two orthogonal projection operators, then let $P_1 \cap P_2$ be the orthogonal projection operator onto $\operatorname{ran}(P_1) \cap \operatorname{ran}(P_2)$.

Before proving Thm. 4, we need to state and prove two lemmas. The first of these appears next.

**Lemma 20.** *Let $K \leq H < \mathcal{O}(m)$ be two finite orthogonal groups such that $|H : K| = 2$. Let $h \in H \setminus K$. Then $P_K \cap P_{2I+h} = P_K - P_H$.*

*Proof.* Since $|H : K| = 2$, then $K \trianglelefteq H$, and hence $H/K = \{K, hK\}$ is a bona fide group. Thus, there exists an isomorphism $\pi : H/K \mapsto \mathbb{Z}_2$, where we define $\mathbb{Z}_2 = \{-1, 1\}$ under multiplication. Since we require $\ker(\pi) = K$, then we have

$$\pi(h) = \begin{cases} 1, & \text{if } h \in K \\ -1, & \text{otherwise.} \end{cases}$$

Now let

$$V = \operatorname{ran}(P_K \cap P_{2I+h}) = \{v \in \mathbb{R}^m : Kv = v, hKv = -v\}.$$

We thus see that $(\pi, V)$ is a representation of $H$, and since $\pi$ is scalar-valued, then $(\pi, V)$ is a direct sum of copies of a single complex-irreducible representation (irrep) of $H$. Noting that $\pi$ is its own complex-irreducible character, we have the orthogonal projection

$$\begin{aligned}
P_K \cap P_{2I+h} &= \frac{1}{|H|} \sum_{h \in H} \pi(h)h \\
&= \frac{1}{|H|} \sum_{g \in K} \pi(g)g + \frac{1}{|H|} \sum_{g \in hK} \pi(g)g \\
&= \frac{1}{|H|} \sum_{g \in K} g - \frac{1}{|H|} \sum_{g \in hK} g \\
&= \frac{1}{|H|} \sum_{g \in K} g - \frac{1}{|H|} \left( \sum_{g \in H} g - \sum_{g \in K} g \right) \\
&= \frac{2}{|H|} \sum_{g \in K} g - \frac{1}{|H|} \sum_{g \in H} g \\
&= \frac{1}{|K|} \sum_{g \in K} g - \frac{1}{|H|} \sum_{g \in H} g \\
&= P_K - P_H.
\end{aligned}$$

$\qquad\square$

The second lemma, appearing below, will be used to characterize the condition $[W_* \mid b_*] \in \Theta_{n_*}$ appearing in Lemma 2.

**Lemma 21.** *Let $J \leq G$ and $\{g_1, \ldots, g_n\}$ a transversal of $G/J$ with $g_1 \in J$. Let $V \leq \operatorname{ran}(P_J)$ be a vector subspace, and let $P_V$ be the orthogonal projection operator onto $V$. Then there exists $w \in V$ such that $g_1 w, \ldots, g_n w$ are distinct vectors if and only if $\operatorname{st}_G(P_V) = J$.*

*Proof.* First we note that if $w \in V$, then $g_1 w, \ldots, g_n w$ are distinct iff $g_i w \neq w \forall i \in \{2, \ldots, n\}$; to see this, we prove the equivalent statement that $g_1 w, \ldots, g_n w$ are not distinct iff $g_i w = w$ for some $i \in \{2, \ldots, n\}$. For the reverse implication, $g_i w = w$ is equivalently $g_i w = g_1 w$, since $g_1 \in J$ and $w \in \operatorname{ran}(P_J)$; $g_1 w$ and $g_i w$ are thus not distinct. For the forward implication, suppose $g_i w = g_j w$ for some distinct $i, j \in \{1, \ldots, n\}$. Then there exists $k \in \{2, \ldots, n\}$ and $g \in J$ such that $g_k g = g_j^{-1} g_i$. We then have

$$g_i w = g_j w$$
$$g_j^{-1} g_i w = w$$
$$g_k g w = w$$
$$g_k w = w.$$

We now prove the stated lemma. Define the vector subspaces

$$V_i = \{v \in V : g_i v = v\} \forall i \in \{2, \ldots, n\}.$$

We have

$$\exists w \in V \mid g_i w \neq w \forall i \in \{2, \ldots, n\} \Leftrightarrow \exists w \in V \mid w \notin V_i \forall i \in \{2, \ldots, n\}$$
$$\Leftrightarrow \exists w \in V \setminus \bigcup_{i=2}^{n} V_i$$
$$\Leftrightarrow V_i < V \forall i \in \{2, \ldots, n\}$$
$$\Leftrightarrow g_i \notin \operatorname{st}(P_V) \forall i \in \{2, \ldots, n\}$$
$$\Leftrightarrow \operatorname{st}(P_V) = J.$$

$\square$

We now prove Thm. 4.

*Proof of Thm. 4.* **(b)** Suppose $\rho_{HK}^{\mathrm{PZ}} \in \operatorname{ran}(F)$. We first prove the forward implication. Suppose $f \in F^{-1}(\rho_{HK}^{\mathrm{PZ}})$. Then the canonical parameters of $f$ satisfy Eqs. 3-6, where the signed perm-rep $\rho$ in Lemma 3 satisfies $\rho \in \rho_{HK}^{\mathrm{PZ}}$. By an appropriate choice of fundamental domain $\Omega_{n_*}$, we can assume without loss of generality that $\rho = \rho_{HK}$. We proceed to prove the claimed expressions for the canonical parameters of $f$.

**Expression for $a_*$:** Regardless of its type, $\rho$ is transitive on $\{e_1, \ldots, e_{n_*}\}$, and thus so is $\pi$. Hence, by Eq. 4, $a_*$ is a constant vector. That $a \neq 0$ follows from the definition of the canonical parameter $a_*$ in Lemma 2.

**Expression for $b_*$:** If $\rho$ is type 1, then it is an irreducible unsigned perm-rep and is transitive on $\{e_1, \ldots, e_{n_*}\}$. Thus, by Eq. 5, $b_*$ is a constant vector. On the other hand, if $\rho$ is type 2, then it is transitive on $\{\pm e_1, \ldots, \pm e_{n_*}\}$. In particular, for every $i = 1, \ldots, n_*$, there exists $g \in G$ such that $\rho(g)e_i = -e_i$. Hence, $b_{*i} = -b_{*i}$ for every $i$, so that $b_* = 0$.

**Expression for $W_*$:** By Prop. 12 (a), the subgroup $K$ satisfies

$$K = \{g \in G : \rho(g)e_1 = e_1\}.$$

Thus, by Eq 3, the first row $w^\top$ of $W_*$ satisfies $w^\top = w^\top g \forall g \in K$, or equivalently $gw = w \forall g \in K$. Thus, $w \in \operatorname{ran}(P_K)$.

In addition, if $\rho$ is type 2, then by Prop. 12a, we have $hw = -w \forall h \in H \setminus K$. Given any choice of $h \in H \setminus K$, we have $hK = H \setminus K$. We thus have

$$hKw = -w$$
$$hw = -w$$
$$(2I + h)w = w.$$

Combining this with $w \in \mathrm{ran}(P_K)$, we have $w \in \mathrm{ran}(P_K \cap P_{2I+h})$. By Lemma 20, we obtain $w \in \mathrm{ran}(P_K - P_H)$. Combining the results for both types 1 and 2, we establish $w \in \mathrm{ran}(P_K - \tau P_H)$. That $\|w\| = 1$ follows from the definition of the canonical parameter $W_*$ in Lemma 2.

Now let $w_1^\top, \ldots, w_{n_*}^\top$ be the rows of $W_*$. Since $G$ and $\rho(G)$ are both orthogonal, then $\rho(g_i^\top)e_i = e_1$, and hence the first row of $\rho(g_i^\top)W_*$ is $w_i^\top$. By Eq. 3, the first row of $W_* g_i^\top$ is $w_i^\top$ as well; thus, since $w_1 = w$, then $w^\top g_i^\top = w_i^\top$, or equivalently $w_i = g_i w$.

**Expression for $c_*$:** We rewrite Eq. 6 as

$$(I - g)c_* = -\frac{a}{2}(I - g)W_* \vec{1} \forall g \in G,$$

where we have used Eq. 8. This is equivalently expressed as

$$(I - P_G)c_* = -\frac{a}{2}(I - P_G)W_* \vec{1}.$$

We focus on the term

$$(I - P_G)W_* \vec{1} = (I - P_G)(g_1 + \ldots + g_n)w.$$

Since $P_G$ is an average over all $g \in G$, then $P_G g_i = P_G \forall i \in \{1, \ldots, n\}$. We thus have

$$(I - P_G)W_* \vec{1} = (g_1 + \ldots + g_n)w - nP_G w.$$

If $\rho$ is type 1 so that $w \in \mathrm{ran}(P_K) = \mathrm{ran}(P_H)$, then

$$(g_1 + \ldots + g_n)w = (g_1 + \ldots + g_n)P_H w$$
$$= nP_G w,$$

so that $(I - P_G)W_* \vec{1} = 0$. On the other hand, if $\rho$ is type 2, then $hw = -w \forall h \in H \setminus K$; thus, $w$ cannot be fixed under all of $G$, so that $P_G w = 0$ and hence

$$(I - P_G)W_* \vec{1} = (g_1 + \ldots + g_n)w = W_* \vec{1}.$$

Combining the results for both types 1 and 2, we have $(I - P_G)W_* \vec{1} = \tau W_1 \vec{1}$ and thus

$$(I - P_G)c_* = -\frac{1}{2}a\tau W_* \vec{1}.$$

Since we already know the right-hand side is in $\mathrm{ran}(I - P_G)$, then we obtain the expression for $c_*$ as claimed.

For the reverse implication, let $f$ be a $G$-SNN whose canonical parameters satisfy Eqs. 7-10. Then it is easy to see that the canonical parameters of $f$ also satisfy Eqs. 3-6. Lemma 3 thus implies $F(f) = \rho_{HK}^{\mathrm{PZ}}$.

**(a)** By part (b) of this theorem, $\rho_{HK}^{\mathrm{PZ}} \in \mathrm{ran}(F)$ iff there exists a $G$-SNN $f$ whose canonical parameters satisfy Eqs. 7-10. Without loss of generality, we assume $a = 1$ in Eq. 8 and $c = 0$ in Eq. 10. Then $\rho_{HK}^{\mathrm{PZ}} \in \mathrm{ran}(F)$ iff there exists $[W_* \mid b_*] \in \Theta_{n_*}$ such that $W_*$ satisfies Eq. 7 and $b_*$ satisfies Eq. 9. We now have two separate cases depending on the type of $\rho_{HK}$.

**Case 1:** Suppose $\rho_{HK}$ is type 1. Without loss of generality, we assume $b \neq 0$ in Eq. 9. Then $[W_* \mid b_*]$ has pairwise nonparallel rows iff $W_*$ has distinct rows. By Eq. 7, $\rho_{HK}^{\mathrm{PZ}} \in \mathrm{ran}(F)$ iff there exists $w \in \mathrm{ran}(P_K)$, such that $g_1 w, \ldots, g_n w$ are distinct; note this $w$ is necessarily nonzero, and hence without loss of generality, we assume $\|w\| = 1$. Noting that $H = K$ since $\rho_{HK}$ is type 1, and invoking Lemma 21 with $J = K$ and $V = \mathrm{ran}(P_K)$, we establish the claim.

**Case 2:** Suppose $\rho_{HK}$ is type 2. Then $b = 0$ in Eq. 9, and $[W_* \mid b_*]$ has pairwise nonparallel rows iff so does $W_*$. Thus, by Eq. 7, $\rho_{HK}^{\mathrm{PZ}} \in \mathrm{ran}(F)$ iff there exists $w \in \mathrm{ran}(P_K - P_H)$ such that $g_1 w, \ldots, g_n w$ are pairwise nonparallel; note this $w$ is necessarily nonzero, and hence without loss of generality we can assume $\|w\| = 1$. For any $h \in H \setminus K$, we have $-g_i w = g_i h w = g_{-i} w \forall i \in \{1, \ldots, n\}$. Moreover, $\{g_{\pm 1}, \ldots, g_{\pm n}\}$ is a transversal of $G/K$. Invoking Lemma 21 with $J = K$ and $V = \mathrm{ran}(P_K - P_H)$, we thus establish the claim. $\square$

Table 1: Ratio of the number of irreducible $G$-SNN architectures to the number of irreducible signed perm-reps of each type (1 vs. 2) for every group $G$, $|G| \leq 8$, up to isomorphism. The particular representations used for each group are described in the main text.

| $G$ | Type 1 | Type 2 | $G$ | Type 1 | Type 2 |
|-----|--------|--------|-----|--------|--------|
| $C_2$ | 2/2 | 1/1 | $\{e\}$ | 1/1 | 0/0 |
| $C_3$ | 2/2 | 0/0 | $C_2^2$ | 4/5 | 3/6 |
| $C_4$ | 3/3 | 2/2 | $C_2^3$ | 8/16 | 7/35 |
| $C_5$ | 2/2 | 0/0 | $C_2 \times C_4$ | 6/8 | 5/11 |
| $C_6$ | 4/4 | 2/2 | $D_3$ | 3/4 | 1/2 |
| $C_7$ | 2/2 | 0/0 | $D_4$ | 5/8 | 7/13 |
| $C_8$ | 4/4 | 3/3 | $Q_8$ | 6/6 | 7/7 |

## C    Examples

### C.1    Irreducible architecture count

Using our code implementation, we enumerated all irreducible $G$-SNN architectures for every group $G$, $|G| \leq 8$, up to isomorphism. For each group, we consider only one particular permutation representation defined as follows: First, let $[i_1, \ldots, i_n]$ denote the permutation on the orthonormal basis $\{e_1, \ldots, e_n\}$ where $e_j \to e_{i_j}$. We then represent the cyclic group $C_n$ by the set of cyclic permutations generated by $[2, \ldots, n, 1]$, and we represent the dihedral group $D_n$ as the group generated by $C_n$ together with the reversing permutation $[n, n-1, \ldots, 1]$. We represent the direct product of groups by the direct sum of the factor groups; e.g., if $G_1$ acts on $[1, \ldots, n_1]$ and $G_2$ acts on $[1, \ldots, n_2]$, then $G_1 \times G_2$ acts on $[1, \ldots, n_1 + n_2]$ with $G_1$ acting on the first $n_1$ elements and $G_2$ acting on the last $n_2$ elements. Finally, we represent the quaternian group $Q_8$ in terms of the following generators:

$$i = [3, 4, 2, 1, 7, 8, 6, 5]$$
$$j = [5, 6, 8, 7, 2, 1, 3, 4]$$
$$k = [7, 8, 5, 6, 4, 3, 2, 1].$$

For each group $G$, we report the ratio of the number of irreducible $G$-SNN architectures of each type to the number of irreducible signed perm reps of the respective type (Table 1). We see that there are generally fewer type 2 architectures—which are the topologically nontrivial ones—than type 1 architectures, although the number of type 2 architectures is not negligible. We also observe that—especially for the direct products of groups—there is a large number of irreducible signed perm reps that do not satisfy the condition in Thm. 4 (a); this is likely because in the rejected architectures, some of the weight vectors are constrained such that the architecture is equivalent to a smaller architecture already enumerated. This trend also motivates the need for more intuition about the condition in Thm. 4 (a).

### C.2    The dihedral permutation group

Consider the dihedral group $G = D_6$ of permutations generated by

$$r = [2, 3, 4, 5, 6, 1] \tag{11}$$
$$t = [6, 5, 4, 3, 2, 1]. \tag{12}$$

There are 14 irreducible $G$-SNN architectures– 7 of each type. We visualize their canonical weight matrices and corresponding cohomology classes in Figs. 4-6. Each architecture is named "i.j" where i and j index the subgroups $H$ and $K$ that are used to construct the architecture (see Thm. 4); Table 2 lists these subgroups for each architecture.

In contrast to the cyclic permutation group (Sec. 4.1), the cohomology class illustrations for $G = D_6$ have arcs of two colors; red (resp. blue) arcs represent the action of the generator $r$ (resp. $t$). The existence of any loops with an odd number of dashed arcs indicates a nontrivial topology. For example, the four architectures 4.j with three hidden neurons correspond to the classes in $\mathcal{H}^1(G, M_{H_4}) \cong C_2 \times C_2$.

Table 2: Subgroups $K_j \leq H_i \leq G$ of the dihedral permutation group $G = D_6$ such that the irreducible signed perm rep $\rho_{H_i K_j}$ admits a corresponding irreducible $G$-SNN architecture named "i.j" in Figs. 4-6. In each row, $|H_i : K_0| = 1$ and $|H_i : K_j| = 2$ for $j \geq 1$. The generators $r$ and $t$ are defined in Eqs. 11-12.

|       | $K_0$            | $K_1$            | $K_2$        | $K_3$           |
|-------|------------------|------------------|--------------|-----------------|
| $H_0$ | $\langle e \rangle$ |                  |              |                 |
| $H_1$ | $\langle r^3 \rangle$ | $\langle e \rangle$ |              |                 |
| $H_2$ | $\langle t \rangle$ | $\langle e \rangle$ |              |                 |
| $H_3$ | $\langle r^3 t \rangle$ | $\langle e \rangle$ |              |                 |
| $H_4$ | $\langle r^3, t \rangle$ | $\langle r^3 \rangle$ | $\langle t \rangle$ | $\langle r^3 t \rangle$ |
| $H_5$ | $\langle r^2, rt \rangle$ |              |              |                 |
| $H_6$ | $D_6$            | $\langle r^2, t \rangle$ |              |                 |

Architecture 0.0

Architecture 5.0

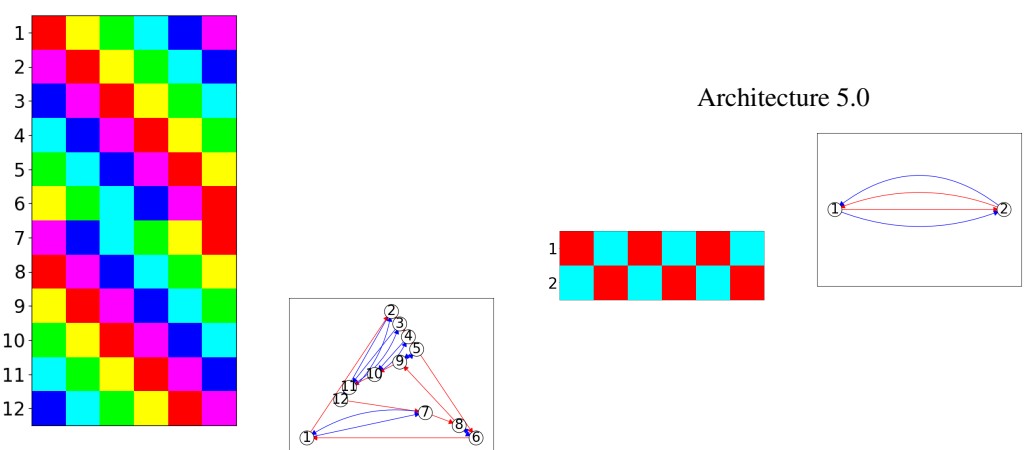

Figure 4: Constraint patterns of the weight matrices and illustrations of the cohomology classes of two irreducible $G$-SNN architectures for the dihedral permutation group $G = D_6$. These are the only two architectures with no partnering type 2 architectures. Interpretation is the same as in Fig. 1. Red (resp. blue) arcs represent the action of the generator $r$ (resp. $t$) of $D_6$ (see Eqs. 11-12). See Table 2 to interpret the names "architecture i.j".

An important remark is that while there are only two architectures 6.j corresponding to the subgroup $H_6$, the corresponding cohomology group is $\mathcal{H}^1(G, M_{H_6}) \cong C_2 \times C_2$. Thus, there are two cohomology classes for which the corresponding signed perm reps failed the condition in Thm. 4 (a). It is thus not necessary for an irreducible architecture to exist for every cohomology class.

We also draw the network morphisms given by asymptotic inclusions between the irreducible architectures, as well as the shortcuts due to topological tunneling (Fig. 7); see Sec. 5 for exposition on these concepts. As with Fig. 3, we determined the asymptotic inclusions manually by looking at the first rows of the weight matrices depicted in Figs. 4-6 and observing how they nest. We obtain a 4-partite topology on the architecture space, plus some topological "tunnels" between architectures belonging to a common cohomology ring.

## C.3 The dihedral rotation group

As our final example, we consider a 2D orthogonal representation of the dihedral group $G = D_6$. In this representation, the generator $r$ is a $60°$ counterclockwise rotation, and the generator $t$ is a reflection about the line $y = \tan(15°)x$; we choose this line of reflection solely because it makes the example interesting. There are six irreducible $G$-SNN architectures for this group—three of each

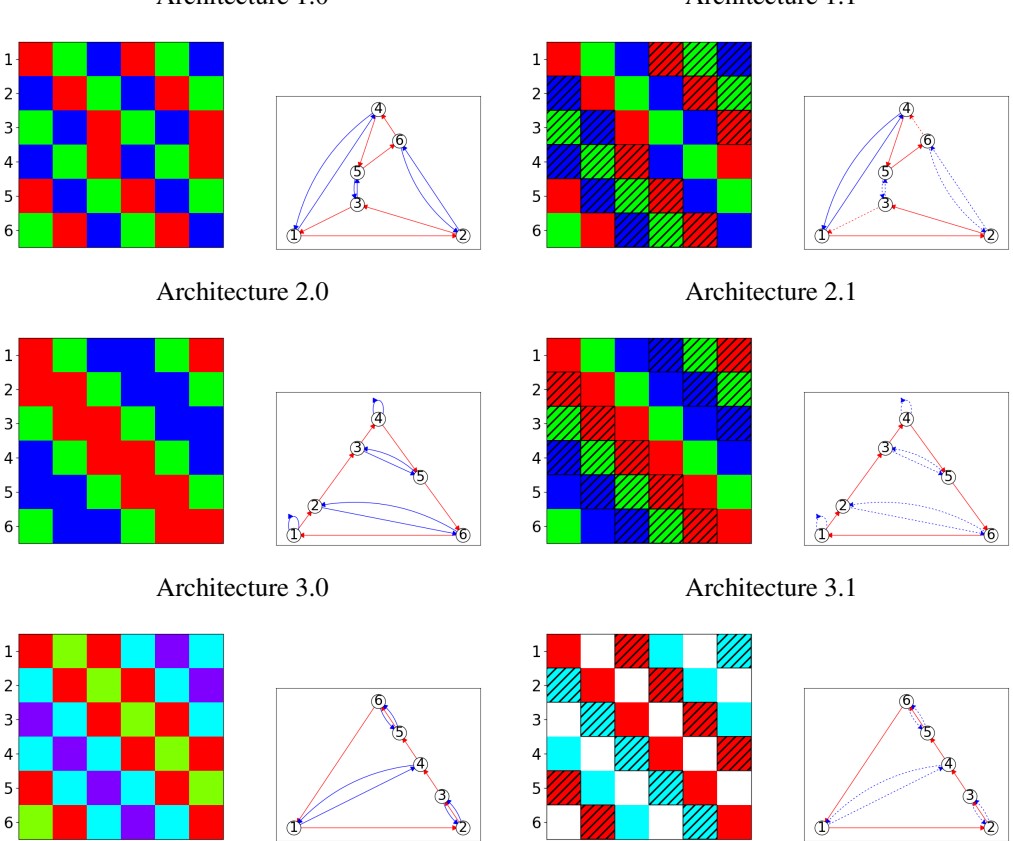

Figure 5: Constraint patterns of the weight matrices and illustrations of the cohomology classes of the six-hidden-neuron irreducible $G$-SNN architectures for the dihedral permutation group $G = D_6$. Interpretation is the same as in Fig. 1. Weights colored white are constrained to equal zero. See Table 2 to interpret the names "architecture i.j".

type—and we visualize there contour plots (Fig. 8). The level curves are clearly invariant under $60°$ rotations and are symmetric about $y = \tan(15°)x$. The architecture names are still based on Table 2, but the generators $r$ and $t$ are now 2D orthogonal transformations instead of permutations. Note that for architectures 0.0 and 1.1, the corresponding subgroup $K$ appearing in Thm. 4 is $K = \langle e \rangle$; for architectures 2.0 and 4.2, $K = \langle t \rangle$; and for architectures 3.0 and 4.3, $K = \langle r^3 t \rangle$, whence the columns in Fig. 8.

Each type 2 architecture in the second row of Fig. 8 is asymptotically included in the type 1 architecture depicted above it. This is the same situation as with the cyclic rotation group in Sec. 4.2; each type 1 architecture approaches the corresponding type 2 architecture below it as its bias parameter $b_*$ tends to zero. In contrast to the cyclic rotation group, however, there are other architectures that are effectively confined from one another. The weight vectors of architecture 2.0 are orthogonal to those of architecture 3.0, and hence one architecture can reach the other only if it passes through a degenerate network with zero-valued weight vectors. By the same token, architectures 4.2 and 4.3 are topologically confined from one another, just as Prop. 6 states.

Finally, we note that while we have architecture 1.1, there is no architecture 1.0; similarly, we have architectures 4.2 and 4.3 but not 4.0 and 4.1. This example thus demonstrates that the cohomology classes in a given cohomology ring for which the corresponding $G$-SNN architecture exists (i.e., satisfies the condition in Thm. 4 (a)) need not form a subgroup, and this raises the question: Are there any discernible patterns in the set of irreducible $G$-SNN architectures (which satisfy Thm. 4 (a)) as we vary $G$? We will investigate this further as part of future work.

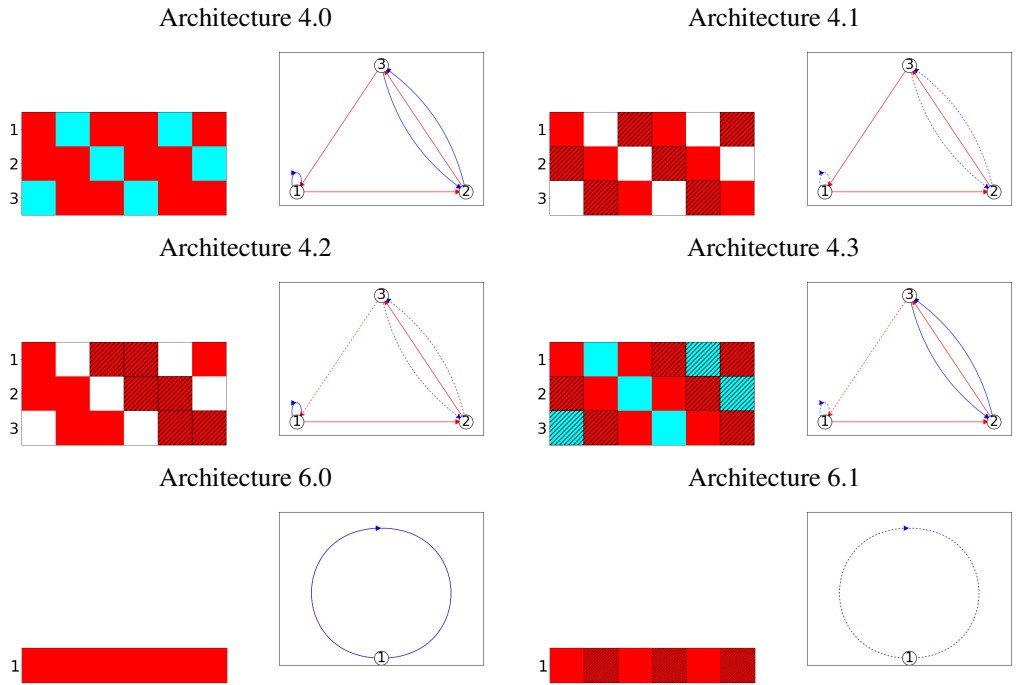

Figure 6: Constraint patterns of the weight matrices and illustrations of the cohomology classes of the three-hidden-neuron and single-hidden-neuron irreducible $G$-SNN architectures for the dihedral permutation group $G = D_6$. Interpretation is the same as in Fig. 1. See Table 2 to interpret the names "architecture i.j".

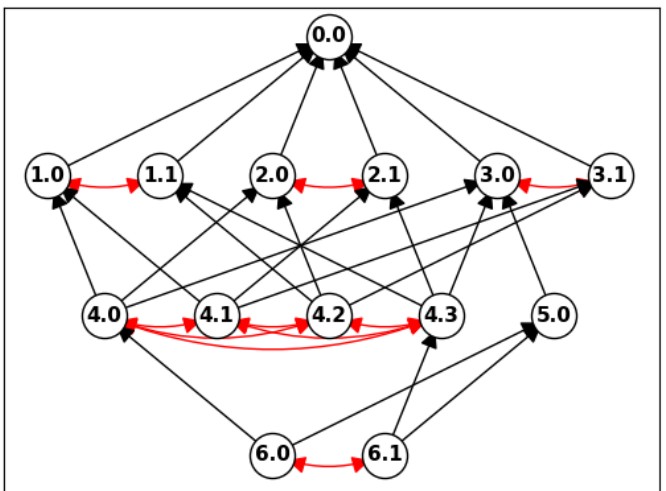

Figure 7: Network morphisms between irreducible $G$-SNN architectures for the dihedral permutation group $G = D_6$. Every direct path in black represents an asymptotic inclusion. Red doubled-arrowed arcs represent the feasibility of topological tunneling.

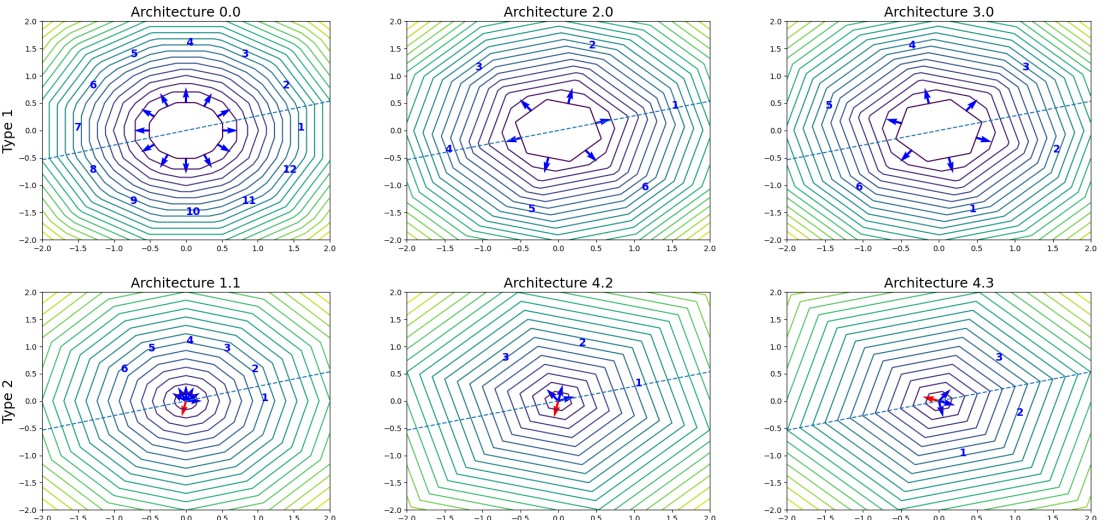

Figure 8: Contour plots of the six irreducible $G$-SNN architectures for the 2D orthogonal representation of $G = D_6$. The architecture names are still based on Table 2, but the generator $r$ is now a $60°$ rotation and $t$ is a reflection about the dashed black line. In architectures 0.0 and 1.1, weight vector 1 is unconstrained and was chosen arbitrarily for visualization. in the other four architectures, however, weight vector 1 is constrained to a 1D subspace as depicted. Interpretation of these plots is the same as in Fig. 2.

## D  Remarks

### D.1  Asymptotic inclusion

Let $\mathrm{SNN}_{\mathrm{irr}}(G)$ be the space of all irreducible $G$-SNNs equipped with the topology of uniform convergence on compact sets. Let $S_+^{m-1}$ denote a hemisphere of the $(m-1)$-dimensional unit sphere such that it has no antipodal pairs of points. Choose an ordering on $G$ so it has elements $g_1, \ldots, g_{|G|}$ with $g_1 = 1$, and define the map $\phi : \mathbb{R}^{\neq 0} \times S_+^{m-1} \times \mathbb{R} \times \mathrm{ran}(P_G) \times \mathbb{R} \mapsto \mathrm{SNN}_{\mathrm{irr}}(G)$ by

$$[\phi(a, w, b, c, d)](x) = a\vec{1}_{|G|}^\top \mathrm{ReLU}(Wx + b\vec{1}_{|G|}) + c^\top x + d \, \forall x \in \mathbb{R}^m,$$

where the $|G| \times m$ weight matrix $W$ is defined as

$$W = \sum_{i=1}^{|G|} e_i(g_i w)^\top.$$

We call this the *unraveled parameterization* of $G$-SNNs, and it has the advantage that it has $|G|$ hidden neurons regardless of the associated signed perm-rep. We can easily transform it into the canonical parameterization as follows: Let $K \leq H \leq G$ be the largest subgroups such that $|H : K| \leq 2$ and $w \in P_K - \tau P_H$ where $\tau = |H : K| - 1$; replace the parameter $a$ with $a|H|$; and finally, use $(a|H|, w, b, c, d)$ and Thm. 4 (b) to construct the canonical form. This is "rolling up" the $G$-SNN so that only $|G|/|H|$ of the $|G|$ hidden neurons remain. Note that this procedure can be reversed, so that we can move back-and-forth between the unraveled and canonical parameterizations. As a consequence, it immediately follows that $\phi$ is a well-defined function, in the sense that it outputs a $G$-SNN that is indeed *irreducible*, and is surjective.

Let $\Omega$ be a fundamental domain in $S_+^{m-1}$ under the action of $G$ (note that $S_+^{m-1} \subset \mathbb{R}^m$), and let $\phi \mid_\Omega$ denote the restriction of $\phi$ to $\mathbb{R}^{\neq 0} \times \Omega \times \mathbb{R} \times \mathrm{ran}(P_G) \times \mathbb{R}$. Then $\phi \mid_\Omega$ is injective as well and thus a bijection; indeed, without this restriction, $w$ and $gw$ for any $g \in G$ would both generate the same weight matrix $W$ in the unraveled parameterization up to the order of its rows, and the restriction to a fundamental domain breaks this redundancy.

The following lemma will help us prove Thm. 5.

**Lemma 22.** *We have:*

(a) $\phi$ *is a continuous function.*

(b) *Let* $\{f_n \in \mathrm{SNN}_{\mathrm{irr}}(G)\}_{n=1}^{\infty}$ *be a sequence such that* $f_n \to f \in \mathrm{SNN}_{\mathrm{irr}}(G)$ *in the topology of* $\mathrm{SNN}_{\mathrm{irr}}(G)$. *Let* $(a_n, w_n, b_n, c_n, d_n) = (\phi \mid_{\Omega})^{-1}(f_n)$ *for each $n$ and similar for $f$. Then there exists $g \in G$ such that* $(gw_n, b_n) \to \pm(w, b)$.

*Proof.* **(a)** Let $\{\theta_n \in \mathrm{dom}(\phi)\}_{n=1}^{\infty}$ be a convergent sequence with limit $\theta \in \mathrm{dom}(\phi)$. For each $n$, let $f_n = \phi(\theta_n)$, and let $f = \phi(\theta)$. For each $x \in \mathbb{R}^m$, the function $\phi(\cdot)(x) : \mathrm{dom}(\phi) \mapsto \mathbb{R}$ is continuous and hence $f_n \to f$ pointwise over the entire domain $\mathbb{R}^m$.

Now since $G$-SNNs are piecewise-linear functions and $\{\theta_n\}_{n=1}^{\infty}$ has a finite limit $\theta$, then clearly the derivatives of the $f_n$ are uniformly bounded, so that in particular $\{f_n\}_{n=1}^{\infty}$ is a sequence of equicontinuous functions. By the Arzelà-Ascoli Theorem, $f_n \to f$ uniformly on every compact set; i.e., $f_n \to f$ in the topology of $\mathrm{SNN}_{\mathrm{irr}}(G)$, thus establishing the continuity of $\phi$.

**(b)** Since $a \neq 0$, then $f$ is nonlinear. For $f_n$ to converge to a nonlinear function, at least one hyperplane on which $f_n$ is non-differentiable must converge to a hyperplane on which $f$ is non-differentiable. Thus, there exist $g_1, g_2 \in G$ such that $(g_1 w_n, b_n) \to \pm(g_2 w, b)$. Equivalently, there exists $g \in G$ such that $(gw_n, b_n) \to \pm(w, b)$. $\qquad\square$

We now prove Thm. 5.

*Proof of Thm. 5.* For the forward implication, suppose $f_2^{\mathrm{PZ}} \hookrightarrow f_1^{\mathrm{PZ}}$. Let $w \in \mathrm{ran}(P_{K_2} - \tau_2 P_{H_2})$ where $\tau_2 = |H_2 : K_2| - 1$. Then there exists $g_2 \in G$ such that $\pm g_2 w \in \mathrm{ran}(P_{K_2} - \tau_2 P_{H_2}) \cap \Omega$; without loss of generality, we assume $g_2 w \in \mathrm{ran}(P_{K_2} - \tau_2 P_{H_2}) \cap \Omega$. Now, there exists $f \in f_2^{\mathrm{PZ}}$ with top weight vector $g_2 w$. Since $f_2^{\mathrm{PZ}} \hookrightarrow f_1^{\mathrm{PZ}}$, then there exists $\{f_n \in f_1^{\mathrm{PZ}}\}_{n=1}^{\infty}$ such that $f_n \to f$ in the topology of $\mathrm{SNN}_{\mathrm{irr}}(G)$. By Lemma 22 (b), there exists $g_1 \in G$ such that $(g_1 w_n, b_n) \to \pm(g_2 w, b)$, where $w_n$ and $b_n$ are the weight and bias parameters of $f_n$ and $b$ is the bias parameter of $f$ respectively. Thus, there exists $g \in G$ such that $\pm g w_n \to w$. Since $w_n \in \mathrm{ran}(P_{K_1} - \tau_1 P_{H_1})$ where $\tau_1 = |H_1 : K_1| - 1$, then $\pm g w_n \in \mathrm{ran}(g P_{H_1} g^{-1} - \tau_1 g P_{H_1} g^{-1})$. Letting $(H, K) = g(H_1, K_1)g^{-1}$, we have $w_n \in \mathrm{ran}(P_K - \tau P_H)$ where $\tau = |H : K| - 1$. We thus establish that $\mathrm{ran}(P_{H_2} - \tau_2 P_{H_2}) \subseteq \mathrm{ran}(P_K - \tau P_H)$.

The space $\mathrm{ran}(P_{K_2} - \tau_2 P_{H_2})$ is thus in particular fixed pointwise by every element of $K$, so that $K \leq \mathrm{st}_G(P_{K_2} - \tau_2 P_{H_2})$. By Thm. 4 (a), however, $\mathrm{st}_G(P_{K_2} - \tau_2 P_{H_2}) = K_2$, so that $K \leq K_2$. In the case $f_1^{\mathrm{PZ}}$ is type 1, we have $H = K \leq K_2 \leq H_2$ and thus $H \cap K_2 = K$, and hence we are done. Suppose instead $f_1^{\mathrm{PZ}}$ is type 2. Then $f_2^{\mathrm{PZ}}$ must be type 2 as well; if it were type 1, then we could set its bias parameter $b$ to be nonzero, and $f_1^{\mathrm{PZ}}$ would be unable to reach it asymptotically as its own bias is constrained to $b = 0$. With both $f_1^{\mathrm{PZ}}$ and $f_2^{\mathrm{PZ}}$ type 2, we have $\mathrm{ran}(P_{K_2} - P_{H_2}) \subseteq \mathrm{ran}(P_K - P_H)$. In particular, for any $h \in H \setminus K$, we must have

$$h(P_{K_2} - P_{H_2}) = -(P_{K_2} - P_{H_2}).$$

However, for the rows of the canonical weight matrix of any $f \in f_2^{\mathrm{PZ}}$ to be pairwise nonparallel, we must have $g(P_{K_2} - P_{H_2}) = -(P_{K_2} - P_{H_2})$ implies $g \in H_2$. Hence, $H \setminus K \subseteq H_2$. Combining this with $K \leq K_2 < H_2$, we obtain $H \leq H_2$. Finally, since $\mathrm{ran}(P_{K_2} - P_{H_2})$ must be fixed under each element of $K_2$ but *not* fixed under each element of $H \setminus K$, then we must have $(H \setminus K) \cap K_2 = \emptyset$, from which we conclude $H \cap K_2 = K$.

For the reverse implication, suppose there exists $(H, K) \in (H_1, K_1)^G$ such that $H \leq H_2$, $K \leq K_2$, and $H \cap K_2 = K$. Without loss of generality, let $(H, K) = (H_1, K_1)$. Let $f \in f_2^{\mathrm{PZ}}$ an $(a, w, b, c, d) = (\phi \mid_{\Omega})^{-1}(f)$. Define the sequence $\{f_n \in f_1^{\mathrm{PZ}}\}_{n=1}^{\infty}$ and $(a_n, w_n, b_n, c_n, d_n) = (\phi \mid_{\Omega})^{-1}(f_n)$, where we set $a_n = a$, $c_n = c$, and $d_n = d$ for all $n$. We want to show the existence of $w_n$ and $b_n$ such that $w_n \to w$ and $b_n \to b$; Lemma 22 (a) will then give us the desired result.

Suppose $f_2^{\mathrm{PZ}}$ is type 1. Since $K_1 \leq K_2 = H_2$ and $H_1 \leq H_2$, then $K_1 \leq H_1 \leq K_2$ and hence $H_1 \cap K_2 = H_1$. On the other hand, since $H_1 \cap K_2 = K_1$, then $H_1 = K_1$ so that $f_1^{\mathrm{PZ}}$ is type 1 as well. In this case, we set $b_n = b$ for all $n$, and we have $w_n \in \mathrm{ran}(P_{K_1})$ and $w \in \mathrm{ran}(P_{K_2}) \subseteq \mathrm{ran}(P_{K_1})$, thus establishing the existence of a sequence $w_n \to w$. On the other hand, suppose $f_2^{\mathrm{PZ}}$ is type 2.

Then $b = 0$, and we set $b_n = 0$ for all $n$ if $f_1^{\text{PZ}}$ is type 2 or $b_n \to 0$ if $f_1^{\text{PZ}}$ is type 1. From the hypotheses, it is easy to verify that

$$\text{ran}(P_{K_2} - P_{H_2}) \subseteq \text{ran}(P_{K_1} - P_{H_1}) \subseteq \text{ran}(P_{K_1}).$$

It follows that regardless of the type of $f_1^{\text{PZ}}$, a sequence $w_n \to w$ exists, thereby establishing the claim. $\qquad\square$

### D.2 Topological confinement

We prove Prop. 6, which states that non-cohomologous irreducible $G$-SNN architectures are in a sense orthogonal.

*Proof of Prop. 6.* For $i = 1, 2$, let $\tau_i = |H : K_i|$. Then by Eq. 7, we have the constraints $w_i \in \text{ran}(P_{K_i} - \tau_i P_H)$. If one of the $K_i$ equals $H$, say $K_1 = H$, then in particular we have the constraints $P_H w_1 = w_1$ and $(P_{K_2} - P_H) w_2 = w_2$. We thus have

$$\begin{aligned}
w_1^\top w_2 &= w_1^\top P_H^\top (P_{K_2} - P_H) w_2 \\
&= w_1^\top P_H (P_{K_2} - P_H) w_2 \\
&= w_1^\top (P_H P_{K_2} - P_H) w_2 \\
&= 0,
\end{aligned}$$

where the last step holds because $K_2 \leq H$ and hence $P_H P_{K_2} = P_H$.

On the other hand, suppose $K_1$ and $K_2$ are both proper subgroups of $H$. Since $P_{K_i} w_i = w_i$, then

$$w_1^\top w_2 = w_1^\top P_{K_1}^\top P_{K_2} w_2.$$

Let $K = K_1 \cap K_2$. It is well-known that because $K_1$ and $K_2$ are distinct index-2 subgroups of $H$, then $K$ is an index-4 subgroup of $H$ and

$$H/K = \{K, k_1 K, k_2 K, hK\},$$

where $k_i \in K_i \setminus K$ for $i = 1, 2$ and $h \in H \setminus (K_1 \cup K_2)$. We thus have

$$\begin{aligned}
P_{K_1}^\top P_{K_2} &= \left( \frac{1}{|K_1|} \sum_{k \in K_1} k \right)^\top \left( \frac{1}{|K_2|} \sum_{k \in K_2} k \right) \\
&= \frac{1}{|K_1|^2} \left( \sum_{k \in K} k + k_1 \sum_{k \in K} k \right)^\top \left( \sum_{k \in K} k + k_2 \sum_{k \in K} k \right) \\
&= \frac{|K|^2}{|K_1|^2} (P_K + k_1 P_K)^\top (P_K + k_2 P_K) \\
&= \frac{1}{4} P_K^\top (I + k_1^\top)(I + k_2) P_K \qquad\qquad = \frac{1}{4} P_K^\top (I + k_1^\top + k_2 + k_1^\top k_2) P_K.
\end{aligned}$$

Since $P_K w_i = w_i$ for both $i = 1, 2$, then we have

$$\begin{aligned}
w_1^\top w_2 &= \frac{1}{4} w_1^\top P_K^\top (I + k_1^\top + k_2 + k_1^\top k_2) P_K w_2 \\
&= \frac{1}{4} w_1^\top (I + k_1^\top + k_2 + k_1^\top k_2) w_2.
\end{aligned}$$

Since $k_1 \in K_1$, then $k_1 w_1 = w_1$, and hence $w_1^\top k_1^\top w_2 = w_1^\top w_2$. On the other hand, since $k_1^\top \in H \setminus K_2$, then $k_1^\top w_2 = -w_2$ so that $w_1^\top k_1^\top w_2 = -w_1^\top w_2$, thus implying $w_1^\top k_1^\top w_2 = 0$. We can similarly show that $w_1^\top k_2 w_2 = 0$. This leaves

$$\begin{aligned}
w_1^\top w_2 &= \frac{1}{4} w_1^\top (I + k_1^\top k_2) w_2 \\
&= \frac{1}{4} (w_1^\top w_2 + w_1^\top w_2) \\
&= \frac{1}{2} w_1^\top w_2,
\end{aligned}$$

implying $w_1^\top w_2 = 0$. $\qquad\square$