# OpenReview forum: "A Classification of $G$-invariant Shallow Neural Networks"
_NeurIPS.cc/2022/Conference — NeurIPS 2022 Accept_

### Official Review · Reviewer_qWKe · 2022-07-04

**Rating:** 4
**Confidence:** 2
**Soundness:** 1 poor
**Presentation:** 1 poor
**Contribution:** 1 poor

**Summary:**

The paper analyses the shallow architectures that are invariant to specific group symmetries.

**Questions:**

NA

**Limitations:**

The authors do not really discuss the limitation of this work.

**Strengths And Weaknesses:**

Strength:
- Identifying and understanding G-invariant networks is an import question

Weaknesses:
- Questionable significance. The paper addresses a very specific setting that I am not sure has any merit. Specifying what specific layers are G-invariant or G-equivariant has been done before, e.g. Maron et al "Invariant and Equivariant Graph Networks". This paper only handles 2-layer networks.
- The paper claims to analyses architectures so it could help find the optimal ones, but it only handles very limited and standard architectures.
- Even in the 2-layer network case the significance of theorem 5 which is the main focus of this paper compared to previous characterizations of invariant networks it is unclear to me what additional benefit does this approach have.
- The paper is not clear
- Considering the reflection symmetry is very unnatural as you need to change the architecture to get an equivalent network. While this complicates the analysis greatly, it has no practical importance as one can always standardize an input dimension $x_i$ to be possitive/negative if the output was invariant to this specific sign.
-If the paper considers symmetries of NN, then a symmetry that is relevant is homogeneity to positive scalars as $a^TReLU(Wx+b)=\frac{1}{c}a^TReLU(cWx+cb)$.


Remarks about the writing:
- In proposition 2 the others talk about $M_H$ as a G-module over $\mathbb{F}_2$, but that isn't well defined. A G-module is any abelian group that G acts on so the meaning of a abelian group over a field isn't clear. It seems from context that the authors consider $M_H$ as the vector field $\mathbb{F}_2$ but they don't define (at least wasn't clear to me) the G-module they worth with.
- In lemma 4 you claim there is a unique perm-rep that gives the canonical equation specific properties, however as this is the canonical  form its shapes shouldn't be influenced by a symmetry working on it.

---

> ### Author Response · Authors · 2022-08-02
> **Response to reviewer qWKe Part 1**
>
> We thank the reviewer for their time and providing their comments. Below we try to respond to each point raised, one-by-one.
>
> **NOTE:** Please see revision/main.pdf in the supplementary material for the revised manuscript (there is content on page 10); the original manuscript is left unchanged. Key revisions are the following:
> 1. Rewrote Sec. 2.3 on group cohomology. Prop. 2 in the original submission is now in the supplementary material.
> 2. Lemma 3, Lemma 4, and Thm. 5 are now numbered Lemma 2, Lemma 3, and Thm. 4 respectively.
> 3. Added Thm. 5, which characterizes the network morphisms linking $G$-SNN architectures in architecture space.
> 4. Sec. 4.3 in the original submission is now Sec. 5.
>
> All numbered references to theorems, lemmas, etc. in our responses are based on our latest revision.
>
> ==========
>
> **1)** Questionable significance. The paper addresses a very specific setting that I am not sure has any merit. Specifying what specific layers are G-invariant or G-equivariant has been done before, e.g. Maron et al "Invariant and Equivariant Graph Networks". This paper only handles 2-layer networks.
>
> **Response:** We would like to clarify that we do not "specify" which layers are $G$-invariant or $G$-equivariant; rather, we start with a $G$-invariant function that is expressible as an SNN and then deduce the necessary structure of the layers.
>
> Previous works (such as those discussed in the introduction) build $G$-invariant architectures by composing $G$-equivariant layers, but it is unknown if this is the **only** way to achieve a $G$-invariant architecture. Kondor & Trivedi (2018) leave it as an open conjecture that the layers of a $G$-invariant architecture are necessarily $G$-equivariant, suggesting that the question is challenging. Our work takes a first step towards this conjecture by answering it in the affirmative for two-layer networks and ReLU activation.
>
> **2)** The paper claims to analyses architectures so it could help find the optimal ones, but it only handles very limited and standard architectures.
>
> **Response:** Firstly, this paper takes only a first step towards the problem of NAS, which is the motivation for a research program beginning with this first paper. Before we can tackle the optimization problem itself, we must first understand the search space-- what are the possible $G$-invariant architectures, and what are the network morphisms that relate them?
>
> It is true that we only consider shallow networks, but as is hopefully apparent from our proofs in the supplementary material, even this restricted case is nontrivial. Moreover, from what we have seen, it is not uncommon in theoretical deep learning research to first consider a simple tractable case, such as shallow networks, and then work to extend the results to more complicated settings later. For example, in the literature, this was the course taken in the study of infinitely wide and overparameterized neural networks.
>
> Finally, we note that the type 2 architectures shown, e.g., in Fig. 1 are in fact non-standard.
>
> **3)** Even in the 2-layer network case the significance of theorem 5 which is the main focus of this paper compared to previous characterizations of invariant networks it is unclear to me what additional benefit does this approach have.
>
> **Response:** Please see our response to (1) above. Again, our objective is to classify _all_ $G$-SNN architectures, while previous works only construct specific examples of such architectures. Our hope is that ultimately, a system that performs NAS over $G$-invariant architectures and implements group theory without relying on the practitioner's intuition, would return superior $G$-invariant architectures; this paper is the theoretical foundation for such a system.

---

> > ### Author Response · Authors · 2022-08-02
> > **Response to reviewer qWKe Part 2**
> >
> > **4)** The paper is not clear and has numerous mistakes that makes checking the validity of its theoretical claims challenging and its findings questionable.
> >
> > **Response:** We would certainly be grateful to the reviewer if they could point us to specific mistakes so we can address them! The reviewer did express their confusion about $H$ and $K$ in Thm. 1 (see (7) below), and in response we made edits to clarify the notation.
> >
> > Regarding clarity, we suspect the most opaque part of the paper is Sec. 2.3, the section on group cohomology. In hindsight, we see that the technical details of group cohomology (or even its definition) are not really necessary for most of the paper; its purpose is primarily to provide the basis for the directed graph visualizations in Fig. 1 as well as to connect our work to a fruitful area of mathematics.
> >
> > To simplify Sec. 2.3, we have moved Prop. 2 into the supplementary material and have completely rewritten Sec. 2.3 to give a high-level overview of group cohomology as it pertains to our paper.
> >
> > **5)** Considering the reflection symmetry is very unnatural as you need to change the architecture to get an equivalent network. While this complicates the analysis greatly, it has no practical importance as one can always standardize an input dimension $x_i$ to be possitive/negative if the output was invariant to this specific sign.
> >
> > **Response:** We think the reviewer is referring to canonical parameterization. As explained in Sec. 3.2, the purpose of writing SNNs in canonical form is that in the canonical parameterization, two SNNs are equal if and only if their parameters are equal. Since a $G$-SNN is invariant to the action of $G$, then its parameters must also remain invariant, thus placing constraints on their forms; this is the content of Lemma 3 (Lemma 4 in the original submission).
> >
> > Pertaining to the reflection redundancies of the weight vectors, we have revised the first paragraph of Sec. 3.1 and give a brief example that we hope clarifies our intention.
> >
> > **6)** If the paper considers symmetries of NN, then a symmetry that is relevant is homogeneity to positive scalars.
> >
> > **Response:** This is indeed a redundancy in ReLU networks (we use the term "redundancy" here and in the paper to contrast it from the symmetry transformations in the group $G$), and this is exactly why we constrain the rows of the weight matrix in Lemma 2 to have unit norm, as this removes this redundancy.
> >
> > **7)** In theorem one the authors state the theorem for general $H, K$ such that $|H:K|\leq 2$, however in proof the authors use lemma 8 that uses a very specific $H, K$ and does not apply on any general $H, K$. Moreover, for such general $H, K$ it is not clear why $G/H$ would even be of size $n$. In the theorem part (b) the authors even state a  unique $H, K)^G$ reinforcing the idea that the theorem was about general $H, K$ and that the authors did not simply forget to state the specific $H, K$ they referred to.
> >
> > **Response:** We see in hindsight that our choice of notation in Thm. 1 may have been a source of confusion, and we apologize for this. We have moved the definition of the $\rho_{HK}$ to appear outside of (right above) Thm. 1 in the revised manuscript to prevent this confusion.
> >
> > To explain, the $H$ and $K$ appearing in the proof of Thm. 1b (where we invoke Lemma 8) refer to the $H$ and $K$ appearing in the statement of part (b) of Thm. 1 and **not** the $H$ and $K$ appearing in the definition of a general $\rho_{HK}$. Theorem 1b asserts the existence of a certain $(H, K)^G$, and in its proof we invoke Lemma 8 to construct specific $H$ and $K$ satisfying the claim.
> >
> > The first part of Thm. 1 before the statement of part (a) was only meant to construct a general $\rho_{HK}$ for any $(H, K)^G$. We have moved this outside the theorem statement so the $H$ and $K$ are not confused with those appearing in Thm. 1b.

---

> > > ### Author Response · Authors · 2022-08-02
> > > **Response to reviewer qWKe Part 3**
> > >
> > > **8)** In proposition 2 the others talk about $M_H$ as a G-module over $F_2$, but that isn't well defined. A G-module is any abelian group that G acts on so the meaning of a abelian group over a field isn't clear. It seems from context that the authors consider $M_H$ as the vector field $F_2$ but they don't define (at least wasn't clear to me) the G-module they worth with.
> > >
> > > **Response:** Every vector space is of course, by definition, an **Abelian group** (of vectors) that is compatible with a scalar field. Our understanding is that a module is a generalization of a vector space where the scalar field is replaced by a scalar ring; in particular, every vector space is a module.
> > >
> > > In our paper, $M_H$ is just $F_2^n$, which is just the $n$-dimensional vector space over the finite field $F_2$. It is thus also a module and an Abelian group (the latter with respect to addition modulo 2). While we could have just used the term "vector space" in place of "module", we stuck with the latter as this seems to be the common practice in group cohomology texts.
> > >
> > > **9)** In lemma 4 you claim there is a unique perm-rep that gives the canonical equation specific properties, however as this is the canonical form its shapes shouldn't be influenced by a symmetry working on it.
> > >
> > > **Response:** We are not entirely sure what the reviewer means by "its shapes shouldn't be influenced by a symmetry working on it"; could they please clarify? In Lemma 3, $W_*$ is $n_* \times m$, $a_*$ and $b_*$ are $n_*$-dimensional, $c_*$ is $m$-dimensional, and $d_*$ is a scalar. Here $m$ is just the input dimension, which is fixed, and $n_*$ is the number of hidden neurons in the canonical form, which is also fixed.
> > >
> > > Perhaps the reviewer is referring to the fact that the number of hidden neurons in a general SNN (not in canonical form) is not unique. For example, a number of hidden neurons could possibly sum up to a linear term, which could then be decomposed into just two hidden neurons. We have revised the first paragraph of Sec. 3.1 and added some text after Lemma 2 that discusses this.
> > >
> > > **10)** The authors do not really discuss the limitation of this work.
> > >
> > > **Response:** We kindly point out that the title of our paper specifies "shallow neural networks", as opposed to deep neural networks, which is perhaps the greatest limitation of our work (although as we say in our response to (2) above, this is a natural starting point and we will extend our work to deep networks in the future). Both in the abstract and introduction, we say that this paper is only the first step towards realizing $G$-invariant NAS, and we also specify that we are only considering finite orthogonal groups. Finally, in all theorems, lemmas, propositions, etc., we state all the required hypotheses.

---

> > > > ### Comment · Reviewer_qWKe · 2022-08-03
> > > > **Initial response to authors response**
> > > >
> > > > There might have been some misunderstanding that I took as errors, I need to re-read the proofs.
> > > >
> > > > - One main issue: You don't say anything in theorem 1 on G so it could be any group, but that group might not have an irreducible representation. You need to assume something on G and it is not clear what exactly you assume. Also, you implicitly assume that $|G/H|=n$ but don't state it anywhere. Please make you assumptions clear so the proof can be varified.
> > > >
> > > > - The issue with the way you use G-module is that there is no (to my knowledge) definition of a G-module over a field. If you mean the specific vector space then say so don't give an general term when you mean a specific object.

---

> > > > > ### Author Response · Authors · 2022-08-03
> > > > > **Followup 1 to reviewer qWKe**
> > > > >
> > > > > We thank the reviewer for their prompt initial response. We respond to their concerns below.
> > > > >
> > > > > **1)** One main issue: You don't say anything in theorem 1 on G so it could be any group, but that group might not have an irreducible representation. You need to assume something on G and it is not clear what exactly you assume.
> > > > >
> > > > > **Response:** We kindly point out that, in the very first sentence of Sec. 2.1 (Preliminaries), we write: "Throughout this paper, let $G$ be a finite group of $m\times m$ orthogonal matrices." We establish other notation, such as $\mathcal{P}(n)$ denoting the group of $n\times n$ permutation matrices for arbitrary $n$, in the same paragraph.
> > > > >
> > > > > We defined the notation $G$ upfront because we use it very frequently throughout the paper including in theorem statements as well as the main text. For example, whenever we say "$G$-SNN", we always mean a finite orthogonal group $G$; we specify this as well in the abstract and introduction.
> > > > >
> > > > > **2)** Also, you implicitly assume that $|G/H|=n$ but don't state it anywhere. Please make you assumptions clear so the proof can be varified.
> > > > >
> > > > > **Response:** We kindly point the reviewer to the text right above the statement of Thm. 1; this text was originally included in the theorem statement, but we moved it outside to address (7) in our last response (please see our response to that for details).
> > > > >
> > > > > Above Thm. 1, we write: "Let $(g_1,\ldots,g_n)$ be a transversal of $G/H$."
> > > > >
> > > > > That is, $n$ is by definition $|G/H|$. Note then, that $n$ is implicitly a function of $H$ as well.
> > > > >
> > > > > **3)** The issue with the way you use G-module is that there is no (to my knowledge) definition of a G-module over a field. If you mean the specific vector space then say so don't give an general term when you mean a specific object.
> > > > >
> > > > > **Response:** Since we only require the addition operation in $F_2^n$, then we kindly ask to forget that it's a vector space. That is, please regard $F_2^n$ **only** as an Abelian group whose elements are the tuples in $\{0, 1\}^n$ under addition modulo $2$. We will make this clear in the next revision; we will use the notation $\{0, 1\}^n$ instead of $F_2^n$ so the reader is not reminded of finite fields.

---

> > > > > > ### Comment · Reviewer_qWKe · 2022-08-04
> > > > > > **Followup 2**
> > > > > >
> > > > > >
> > > > > > - Regarding G: Maybe I am a bit rusty on my representation theory but I don't see anything that implies that $G$ acts in a transitive fashion on $\{e_i\}$ when you describe it. If you assume it, please state so explicitly. If not, I don't see how theorem 1 holds.
> > > > > >
> > > > > > - I figured out the source of my main confusion regarding $\{g_1,..,g_n\}$: I thought the dimension was $n$ not $m$ so using $|G/H|=n$ was a strong assumption. However, you use $n$ twice in different meanings and should change. You use $n$ as the size of $G/H$ but also as the degree of the representation.

---

> > > > > > > ### Author Response · Authors · 2022-08-04
> > > > > > > **Followup 3 to reviewer qWKe**
> > > > > > >
> > > > > > > We thank the reviewer for their engagement.
> > > > > > >
> > > > > > > **1)** Regarding G: Maybe I am a bit rusty on my representation theory but I don't see anything that implies that $G$ acts in a transitive fashion on $e_i$ when you describe it. If you assume it, please state so explicitly. If not, I don't see how theorem 1 holds.
> > > > > > >
> > > > > > > **Response:** Is the reviewer referring to the following sentence in the proof of Thm. 1a: "Since $G$ acts transitively on $G/H$ . . . "?
> > > > > > >
> > > > > > > Firstly, every group $G$ acts transitively on itself (this is, in fact, true for **all** groups). To see this, let $g_1g_2\in G$. Then observe that the element $g = g_2g_1^{-1}$ clearly satisfies $gg_1 = g_2$.
> > > > > > >
> > > > > > > By the same token, $G$ acts transitively on $G/H$ for every subgroup $H$; again, this is a general fact about groups. To see this, consider two cosets $g_iH$ and $g_jH$ in $G/H$. Then the element $g=g_jg_i^{-1}$ clearly satisfies
> > > > > > > $$g(g_iH) = (gg_i)H = g_jH.$$
> > > > > > >
> > > > > > > If the reviewer was referring to a different part of the proof, could they maybe point us to the specific sentence so we can address their concern more accurately?
> > > > > > >
> > > > > > > **2)** You use $n$ twice in different meanings and should change. You use $n$ as the size of  $G/H$ but also as the degree of the representation.
> > > > > > >
> > > > > > > **Response:** There is a reason for this. The elements of $G$ are indeed $m\times m$ matrices. But when we consider the action of $G$ on $G/H$ (as described in our response to (1) above), then each $g\in G$ effectively acts as a permutation of degree $n$ because it permutes the $n$ cosets in $G/H$.
> > > > > > >
> > > > > > > The reason why $\rho_{HK}$ has degree $n$ is because it is defined in terms of the action of $G$ on $G/H$. In fact, in the type 1 case (where the subgroups $H$ and $K$ are equal), $\rho_{HK}$ is **equivalent** to the action of $G$ on $G/H$; if $g$ sends $g_iH$ to $g_jH$, then we say $\rho_{HK}(g)$ sends $e_i$ to $e_j$ (again, in the case $H=K$).
> > > > > > >
> > > > > > > Note that in the most extreme case $H=K=G$, $G/H$ contains just one coset, and hence every element of $G$ acts as the identity permutation on this one element. Correspondingly, $\rho_{HK}$ is just the trivial representation, which is degree $1$.
> > > > > > >
> > > > > > > Note: The case $H\neq K$ is a bit more complicated, but the point remains the same.

---

> > > > > > > > ### Author Response · Authors · 2022-08-05
> > > > > > > > **Followup 4 to reviewer qWKe**
> > > > > > > >
> > > > > > > > We append the following to point (1) in our initial response:
> > > > > > > >
> > > > > > > > We thank the reviewer for bringing to our attention the paper "Invariant and equivariant graph networks" (Maron 2019). We have read through the paper, and we would like to point out that our work is very different from Maron et al. Maron et al classify all $G$-invariant and $G$-equivariant **linear** layers. In contrast, in our work we only assume the SNN to be $G$-invariant as a whole, and we make no assumptions on the linear layers; we then proceed to deduce the network structure necessary for $G$-invariance.
> > > > > > > >
> > > > > > > > Maron et al is nevertheless an important paper for understanding possible $G$-invariant network architectures (building upon the works of e.g. Kondor & Trivedi, 2018), and it was our oversight to not cite it; we will cite it in our next revision.

---

> > > > > > > > > ### Comment · Reviewer_qWKe · 2022-08-08
> > > > > > > > > **Followup 5**
> > > > > > > > >
> > > > > > > > > Reading again I see that there where a few major points I misunderstood so I fixed my score.
> > > > > > > > >
> > > > > > > > > I do think the novelty and contributions are small. The difference between linear layers and your exploration of shallow architecture is small, as there is one main matrix of importance. The biggest difference is that the analysis of layers is scalable while I find it hard to see this approach scaling to larger networks.
> > > > > > > > >
> > > > > > > > > In the end, it is simply taking the action on the feature space and splitting it to orbits. I think you could get this result in a much simpler fashion.

---

> > > > > > > > > > ### Author Response · Authors · 2022-08-08
> > > > > > > > > > **Followup 6 to reviewer qWKe**
> > > > > > > > > >
> > > > > > > > > > We thank the reviewer profusely for their engagement, open-mindedness, and reconsideration of our work!
> > > > > > > > > >
> > > > > > > > > > Below we kindly respond to the reviewer's comments and make some clarifications regarding the novelty and contributions of our work.
> > > > > > > > > >
> > > > > > > > > > **1)** I do think the novelty and contributions are small. The difference between linear layers and your exploration of shallow architecture is small, as there is one main matrix of importance.
> > > > > > > > > >
> > > > > > > > > > **Response:** We respectfully disagree that the leap from $G$-equivariant linear layers to $G$-SNNs is small, as we have identified a significant number of architectures (over twice as many, for some groups $G$) not accounted for by previous works; we explain below.
> > > > > > > > > >
> > > > > > > > > > Maron et al (2019) only consider the case where both the input and output of a linear layer transform by **unsigned** perm-reps. Cohen et al (2019) consider $G$-equivariant linear layers that transform by more general reps of $G$, but they also require the nonlinearity layer to be $G$-equivariant and state that this places restrictions on the reps by which the linear layers can transform. Moreover, they do not identify these admissible reps for the case of ReLU; presumably, according to Cohen et al (2016), only **unsigned** perm-reps are used together with elementwise nonlinearities like the ReLU.
> > > > > > > > > >
> > > > > > > > > > In contrast to Maron et al (2019), our work holds for any finite orthogonal group $G$. Moreover, in contrast to all the works cited above, we **deduce** that if using the ReLU nonlinearity, then the first linear layer can transform by a **signed** perm-rep of $G$. This means there are more $G$-SNN architectures than just those whose first linear layer transforms by **unsigned** perm-reps. For example, please see architectures 1.1 and 3.1 in Fig. 1; notice how each row of the weight vector is a shift and **reflection** of the row above it.
> > > > > > > > > >
> > > > > > > > > > To get an idea of just how many $G$-SNN architectures previous works did not identify, please see Table 1 in the supplementary material (Supp. C.1). For example, for the dihedral group $G=D_4$, previous works would have identified at most 5 irreducible $G$-SNN architectures which correspond to **unsigned** perm-reps acting on the weight vectors; however, there are **7** additional architectures corresponding to proper **signed** perm-reps. We have therefore identified over twice as many architectures as previous works, for shallow networks alone.
> > > > > > > > > >
> > > > > > > > > > If it helps, when the first linear layer transforms by a signed perm-rep of $G$, then the post-ReLU activations transform by (1) a permutation matrix and (2) the addition of $G$-equivariant skip connections; it is these skip connections that previous works did not consider (see first paragraph of Sec. 3.1).
> > > > > > > > > >
> > > > > > > > > > Finally, Prop. 6 clarifies the importance of these additional nonstandard architectures, as it implies that there are datasets that $G$-SNNs cannot fit to, unless they utilize these nonstandard architectures.
> > > > > > > > > >
> > > > > > > > > > **2)** The biggest difference is that the analysis of layers is scalable while I find it hard to see this approach scaling to larger networks.
> > > > > > > > > >
> > > > > > > > > > **Response:** Firstly, as discussed above, the scalability of the layer-by-layer approach of previous works comes at the cost of missing a large number of architectures, namely the ones corresponding to proper **signed** perm-reps. Given that we have already identified numerous nonstandard $G$-SNN architectures, we expect to find many more in the deep case.
> > > > > > > > > >
> > > > > > > > > > We are currently working on extending our results to deep networks, but we felt that we should not delay in sharing our results on shallow networks with the research community and that the deep result is beyond the scope of a first paper. Again, while it is challenging, we think the payoff in the form of new $G$-invariant architectures as well as a complete theoretical understanding of the classification is worth the effort. Our starting point is that we know how the post-ReLU activations transform, and thus we can study how the second layer must be equivariant to this. While this is a bottom-up approach, we believe it will give insight into the ultimate top-down classification.
> > > > > > > > > >
> > > > > > > > > > We would also like to comment that from what we have seen, it is not uncommon in theoretical deep learning research to first consider a simple tractable case, such as shallow networks, and then work to extend the results to more complicated settings later. For example, in the literature, this was the course taken in the study of infinitely wide and overparameterized neural networks.

---

> > > > > > > > > > > ### Author Response · Authors · 2022-08-08
> > > > > > > > > > > **Followup 7 to reviewer qWKe**
> > > > > > > > > > >
> > > > > > > > > > > **3)** In the end, it is simply taking the action on the feature space and splitting it to orbits. I think you could get this result in a much simpler fashion.
> > > > > > > > > > >
> > > > > > > > > > > **Response:** The question is, what reps can we use for this action? The answer is not just perm-reps but **signed** perm-reps, which is one result of our work. Furthermore, we think the condition $st_G(P_K-\tau P_H)=K$ in Thm. 4a is nontrivial; this ccondition is necessary to avoid the enumeration of degenerate architectures or two architectures that turn out to be equivalent.
> > > > > > > > > > >
> > > > > > > > > > > Regarding a simpler proof, we have tried to find a more direct proof but have not yet succeeded. We think what makes the proof long is its "if and only if" nature; i.e., we must guarantee that **every** irreducible  $G$-SNN architecture is enumerated and **exactly once**.
> > > > > > > > > > >
> > > > > > > > > > > Finally, our contribution goes beyond the classification of $G$-SNN architectures, as we now also characterize the network morphisms linking them in architecture space (see Thm. 5 in the revision).

---

### Official Review · Reviewer_FsV1 · 2022-07-07

**Rating:** 7
**Confidence:** 3
**Soundness:** 4 excellent
**Presentation:** 3 good
**Contribution:** 3 good

**Summary:**

The paper gives a classification of all shallow (one hidden layer) ReLU networks that are G-invariant, i.e. invariant under an operation of a an element of a group G on the input. The main result of the paper is a classification theorem, that allows the authors, given some finite orthogonal group G, to classify all the "unique" shallow ReLU network architectures. That is, using the results in the paper one can find all the possible network architecture and weight sharing "schemes". To my understanding, this is the first work to achieve such result.

**Questions:**

- How does the number of neurons grow with the size (or "complexity") of the group G?
- What is the computational cost of running the enumeration algorithm? It seems that this algorithm was run on small groups, is it practical to run on larger group, or does the run-time scale badly with the group size?
- Is the sorting of architectures via the asymptotic inclusion applicable for every group?

**Limitations:**

Yes

**Strengths And Weaknesses:**

The paper provides a novel result that improves the fundamental understanding of group invariant networks. While group invariant networks have been well-studied in previous literature, the result in the paper enables a more detailed understanding of such invariant network, by providing a complete classification of shallow ReLU networks invariant under the operation of a finite orthogonal group. Using the main theorem, the authors can compute all the invariant network architectures given such group. These result open up interesting questions for further research, such as extension of the results to deep networks and using the technical tools developed in the paper for improving neural network architecture design in practice.

While the mathematical contribution is sound, interesting and novel, I believe that the discussion on the implication of the results lacks clear and rigorous statements. It is not clear in what cases the sorting of the G-SNN architectures via the asymptotic inclusion can be achieved, or how exactly does this improve neural architecture search. It could be helpful to discuss in more quantitative detail what, potentially, are the computational benefit of the classification introduced in the paper to NAS. The topological tunneling section could also benefit from more rigorous statements. It is claimed that the results in the paper imply some failure cases due to bad initialization. What exactly is the chance of failure? Having a weights matrix with any weight-sharing has probability zero, so how can this fail due to initialization?

Another issue I have is with the statement of Lemma 3. It seems to introduce a unique parameterization of neural networks, but I believe it lacks some statement saying that all "equivalent" networks have the same paremeterization.

---

> ### Author Response · Authors · 2022-08-02
> **Response to reviewer FsV1 Part 1**
>
> We thank the reviewer for their helpful comments and suggestions. Below we try to respond to each point raised, one-by-one.
>
> **NOTE:** Please see revision/main.pdf in the supplementary material for the revised manuscript (there is content on page 10); the original manuscript is left unchanged. Key revisions are the following:
> 1. Rewrote Sec. 2.3 on group cohomology. Prop. 2 in the original submission is now in the supplementary material.
> 2. Lemma 3, Lemma 4, and Thm. 5 are now numbered Lemma 2, Lemma 3, and Thm. 4 respectively.
> 3. Added Thm. 5, which characterizes the network morphisms linking $G$-SNN architectures in architecture space.
> 4. Sec. 4.3 in the original submission is now Sec. 5.
>
> All numbered references to theorems, lemmas, etc. in our responses are based on our latest revision.
>
> ==========
>
> **W1.** While the mathematical contribution is sound, interesting and novel, I believe that the discussion on the implication of the results lacks clear and rigorous statements. It is not clear in what cases the sorting of the G-SNN architectures via the asymptotic inclusion can be achieved, or how exactly does this improve neural architecture search. It could be helpful to discuss in more quantitative detail what, potentially, are the computational benefit of the classification introduced in the paper to NAS.
>
> **Response:** In our revised manuscript (Sec. 5.2), we have made the definition of asymptotic inclusion more rigorous. Also, we have added Thm. 5, which gives a group-theoretic characterization of the asymptotic inclusions (and hence network morphisms) between the $G$-SNN architectures. Theorem 5 can thus be used to enumerate the network morphisms algorithmicly. The proof of Thm. 5 is based on a certain non-canonical parameterization of $G$-SNNs, the Arzela-Ascoli Theorem, and Thm. 4a.
>
> As a corollary to Thm. 5, there can be no "isolated" $G$-SNN architectures, in the sense that for every architecture there is a network morphism connecting it to another.
>
> Regarding the utility of this theory, the goal is to figure out how to even do NAS properly over the space of $G$-invariant architectures. Theorem 4 tells us what the possible $G$-SNN architectures are, and Thm. 5 tells us about the possible moves between them during the search. The longterm vision is if a practitioner has a $G$-invariant problem, they could just feed their dataset and the group $G$ into a system that would then perform NAS for them; the hope is that this would result in a $G$-invariant architecture better than whatever the practitioner could have intuited. Indeed, a manual search would be tedious as it would require some knowledge of group theory, and we think there would be a good chance that many of the "less obvious" architectures difficult to intuit would be overlooked.
>
> A rigorous experimental study of NAS for $G$-invariant problems based on our theory, compared to architectures easy to intuit, would test our hypothesis. We think such a study, however, is beyond the scope of this first paper.
>
> **W2.** The topological tunneling section could also benefit from more rigorous statements. It is claimed that the results in the paper imply some failure cases due to bad initialization. What exactly is the chance of failure? Having a weights matrix with any weight-sharing has probability zero, so how can this fail due to initialization?
>
> **Response:** We see now that our discussion of initialization was quite unclear. We have removed the initialization example from the topological tunneling section and replaced it with a more sound discussion that makes the same point.
>
> Our point was only that two architectures of the same size can still be unreachable from one another if they have incompatible (or in the examples we give, orthogonal) weight-sharing patterns. Topological tunneling lets us transform one weight-sharing pattern into another in a way analogous to cutting and regluing a Mobius band to remove the twist.
>
> Also, Prop. 6 is the theoretical foundation of "topological confinement", and we now state it directly in the topological tunneling section.
>
> **W3.** Another issue I have is with the statement of Lemma 3. It seems to introduce a unique parameterization of neural networks, but I believe it lacks some statement saying that all "equivalent" networks have the same paremeterization.
>
> **Response:** We are not sure if we fully grasp the reviewer's comment here, and we kindly ask for clarification.
>
> Lemma 2 says that if we are given a function $f$ that is, in principle, expressible as an SNN, then in the canonical parameterization, there is exactly one SNN that expresses $f$.
>
> Consequently, if $f_1$ and $f_2$ are two such functions expressible as SNNs, then $f_1(x) = f_2(x)$ for all inputs $x$ if and only if their canonical parameters are equal. Is this what the reviewer is referring to? Would they like us to clarify or emphasize this point in the text?

---

> > ### Author Response · Authors · 2022-08-02
> > **Response to reviewer FsV1 Part 2**
> >
> > **Q1.** How does the number of neurons grow with the size (or "complexity") of the group G?
> >
> > **Response:** The number of hidden neurons of an irreducible $G$-SNN in canonical form depends on the subgroup $H$ used to construct the corresponding signed perm-rep of $G$. Specifically, the $G$-SNN has $n = |G|/|H|$ hidden neurons, and depending on the choice of $H$, $n$ can range between $1$ ($H=G$) and $n=|G|$ ($H$ the trivial group).
> >
> > **Q2.** What is the computational cost of running the enumeration algorithm? It seems that this algorithm was run on small groups, is it practical to run on larger group, or does the run-time scale badly with the group size?
> >
> > **Response:** Each group considered in the paper (every group of order 8 or smaller, up to isomorphism, as well as $D_6$, the dihedral group with 12 elements) ran in under 1.5 s. We list some more execution times for various cyclic groups $C_n$ and dihedral groups $D_n$ below:
> >
> > $C_{64}$: 1.5 s
> > $C_{128}$: 2.3 s
> > $C_{256}$: 10.0 s
> > $C_{512}$: 67.9 s
> > $C_{1024}$: 568.2 s
> >
> > $D_{32}$: 1.6 s
> > $D_{64}$: 2.3 s
> > $D_{128}$: 5.5 s
> > $D_{256}$: 31.7 s
> > $D_{512}$: 233.4 s
> >
> > We could certainly do this analysis and consider other groups such as direct products of cyclic groups which could be relevant to computer vision. However, there are three caveats to such an analysis:
> >
> > First, our code implementation is not optimized for speed, as we were primarily interested in gaining intuition for different groups, even if they're small. For example, our code tries to find coset representatives, sorted in the right order, such that weight-sharing patterns such as those shown in Fig. 1 are apparent.
> >
> > Second, even if we optimized our code, it is probably not the most efficient approach for real-world applications because it works on arbitrary groups and thus does not exploit the structure of well-studied families of groups such as cyclic groups. For example, we suspect that for certain common families of groups, we could work out the exact form of the projection operator $P_K-\tau P_H$ appearing in Thm. 4a, in which case we could circumvent much of the computation, or at least find faster implementations.
> >
> > Moreover, as part of future work, we plan to answer the following question: If $G$ is either the direct product or a semidirect product of two groups $G_1$ and $G_2$, and if we've already classified the $G_1$-SNNs and $G_2$-SNNs, then is there an easy way to immediately write down the $G$-SNNs? That is, could we develop an "algebra" of group-invariant architectures? If we could do this, then this would provide another shortcut to enumerate the $G$-SNNs for more complicated groups $G$.
> >
> > Third and finally, at least for the application of NAS, we would probably never enumerate **all** $G$-SNN architectures at one time. For example, in a simple local search approach, we would start with a medium-sized network (by selecting subgroups $H$ and $K$ of appropriate size), and then we would only enumerate the architectures linked to the initial one via network morphisms, using Thm. 6.
> >
> > **Q3.** Is the sorting of architectures via the asymptotic inclusion applicable for every group?
> >
> > **Response:** Yes, at least for finite orthogonal groups. This sorting can be done algorithmicly using Thm. 5. Please see our response to W1 above for more.

---

### Official Review · Reviewer_XXff · 2022-07-10

**Rating:** 7
**Confidence:** 4
**Soundness:** 4 excellent
**Presentation:** 4 excellent
**Contribution:** 4 excellent

**Summary:**

While equivariance has been shown to be a useful principle for neural network design, there exist many different equivariant architectures in the literature.
This paper provides a first step to answer the question "what are, in practice, the best design choices?"
1) by decomposing any G-equivariant depth-2 ReLU network as a sum of irreducible G-equivariant networks
2) and by classifying all equivalence classes of such irreducible networks.
The paper also includes a method to find such irreducible networks and practical ideas to leverage them in a Neural-Architecture-Search (NAS) context.


**Questions:**

In the second part of the paper, the authors propose leveraging the irreducible networks to build larger model in NAS.
By cleverly using "topological tunneling", one can avoid resorting to larger architectures (i.e. full GCNN models, which use the regular representation of G).
I am curious if similar insights can be used for the inverse process in a "pruning" context, i.e. starting from a large pre-trained GCNN architecture and, then, pruning redundant channels (i.e. zero/redundant irreducible sub-networks) within the regular representations.


Is it correct that the representation in Theorem 1 is the induced representation from H to G of a 1-dimensional representation of H?
This might be useful to mention.

Line 153-154: it is not immeditely clear what construction (with the additional two neurons) the authors have in mind. This becomes clearer later after Lemma 3 (although it's never explained the two neurons are necessary to avoid the ReLU non-linearity). I would recommend spending a few more mords earlier to clarify this construction.

Line 203: Theorem 5a: $\tau$ seems to be an integer in $\{0, 1\}$ while $P_H$ and $P_K$ are sets of matrices. Is that correct? Then, isn't a scalar factor multiplied to $P_A$ irrelevant in the definition of $\text{st}_G$?

Lines 286-290: the statement about the additional neurons making it harder to discover symmetry and to generalize seems in contrast with the recent understanding about wide neural networks, where overparameterization is considered useful for optimization and generalization.

Line 293: Shouldn't $f^{PZ}_2$ be asymptotically included in $f^{PZ}_1$?

Line 343: I am not sure I understood the sentence "any patters in G-SNN architecture space are unclear". Could you clarify or elaborate further on it?



**Limitations:**

The method proposed is limited to 1-hidden layer ReLU networks.
As the authors clearly stated, this provides only a first step in answering the original question.
However, the analysis provided applies to individual layers in a deep network and, therefore, can still turn out useful for NAS of deeper models.




**Strengths And Weaknesses:**

The problem tackled is relevant and well motivated.
The idea is novel and can be of practical interest in the context of NAS, as argued by the authors.

The presentation is clear, although the assumption that the reader is familiar with group cohomology makes the manuscript less accessible.
For instance, I think the authors could include some additional comments after Proposition 2, trying to provide some more intuitions about its meaning.

While it would be interesting to see an implementation of the NAS ideas proposed here (maybe, in a simple experimental setting), I think the theoretical work is a sufficient contribution.

---

> ### Author Response · Authors · 2022-08-02
> **Response to reviewer XXff Part 1**
>
> We thank the reviewer for their helpful comments and suggestions. Below we try to respond to each point raised, one-by-one.
>
> **NOTE:** Please see revision/main.pdf in the supplementary material for the revised manuscript (there is content on page 10); the original manuscript is left unchanged. Key revisions are the following:
> 1. Rewrote Sec. 2.3 on group cohomology. Prop. 2 in the original submission is now in the supplementary material.
> 2. Lemma 3, Lemma 4, and Thm. 5 are now numbered Lemma 2, Lemma 3, and Thm. 4 respectively.
> 3. Added Thm. 5, which characterizes the network morphisms linking $G$-SNN architectures in architecture space.
> 4. Sec. 4.3 in the original submission is now Sec. 5.
>
> All numbered references to theorems, lemmas, etc. in our responses are based on our latest revision.
>
> ==========
>
> **W1.** The presentation is clear, although the assumption that the reader is familiar with group cohomology makes the manuscript less accessible. For instance, I think the authors could include some additional comments after Proposition 2, trying to provide some more intuitions about its meaning.
>
> **Response:** In hindsight, we see that the technical details of group cohomology (or even its definition) are not really necessary for most of the paper; its purpose is primarily to provide the basis for the directed graph visualizations in Fig. 1 as well as to connect our work to a fruitful area of mathematics.
>
> To simplify Sec. 2.3, we have moved Prop. 2 into the supplementary material and have completely rewritten Sec. 2.3 to give a high-level overview of group cohomology as it pertains to our paper. We no longer state in the first footnote that we assume some familiarity with group cohomology.
>
> **W2.** While it would be interesting to see an implementation of the NAS ideas proposed here (maybe, in a simple experimental setting), I think the theoretical work is a sufficient contribution.
>
> **Response:** A rigorous experimental study is beyond the scope of this first paper; however, we have further developed the theory that could help us with such a study.
>
> In Sec. 5.2, we have made the definition of asymptotic inclusions more rigorous, and we have added Thm. 5, which gives a group-theoretic characterization of these asymptotic inclusions (and hence network morphisms) between $G$-SNN architectures. Thm. 5 taken together with Thm. 4 thus tells us the complete structure of $G$-SNN architecture space.
>
> **Q1.** In the second part of the paper, the authors propose leveraging the irreducible networks to build larger model in NAS. By cleverly using "topological tunneling", one can avoid resorting to larger architectures (i.e. full GCNN models, which use the regular representation of G). I am curious if similar insights can be used for the inverse process in a "pruning" context, i.e. starting from a large pre-trained GCNN architecture and, then, pruning redundant channels (i.e. zero/redundant irreducible sub-networks) within the regular representations.
>
> **Response:** This is a clever idea, and we think it should indeed be possible if we are given a **reducible** pretrained $G$-SNN. Writing the $G$-SNN in canonical form, we could decompose it into a sum of irreducible $G$-SNNs, and we could then discard the negligible components. The resulting pruned network will still be **exactly** $G$-invariant.
>
> **Q2.** Is it correct that the representation in Theorem 1 is the induced representation from H to G of a 1-dimensional representation of H? This might be useful to mention.
>
> **Response:** This is indeed correct, and we thank the reviewer for noting it! We have added this observation at the end of Sec. 2.2.
>
> **Q3.** It is not immeditely clear what construction (with the additional two neurons) the authors have in mind. This becomes clearer later after Lemma 3 (although it's never explained the two neurons are necessary to avoid the ReLU non-linearity). I would recommend spending a few more mords earlier to clarify this construction.
>
> **Response:** We have revised the first paragraph of Sec. 3.1 to clarify the redundancy that can arise from the (anti)alignment of hidden neurons, and state a simple example as well. We have also added additional text right after Lemma 3 to explain where the term $c_*^Tx+d_*$ comes from in the canonical form.

---

> > ### Author Response · Authors · 2022-08-02
> > **Response to reviewer XXff Part 2**
> >
> > **Q4.** Theorem 5a: $\tau$ seems to be an integer in $0, 1$ while $P_H$ and $P_K$ are sets of matrices. Is that correct? Then, isn't a scalar factor multiplied to $P_A$ irrelevant in the definition of $st_G$?
> >
> > **Response:** This was a typo that we unfortunately noticed too late. It should read $st_G(P_K-\tau P_H) = K$. We have corrected this in the revision. Also, $P_K$ (and similar for $P_H$) is an orthogonal projection matrix whose image is fixed pointwise under every element in $K$.
> >
> > **Q5.** The statement about the additional neurons making it harder to discover symmetry and to generalize seems in contrast with the recent understanding about wide neural networks, where overparameterization is considered useful for optimization and generalization.
> >
> > **Response:** We have softened our language in Sec. 5.1 (originally Sec. 4.3 Par. 1) to avoid this confusion. However, we would like to clarify here our point:
> >
> > In the hypothetical scenario of a $G$-invariant target function where $G=C_6$, we only suggest that an SNN with $3k+1$ hidden neurons will do no better than one with $3k$ hidden neurons, as at least one of the $3k+1$ hidden neurons will have to zero out anyway. Even $3k+2$ or $3(k+1)$ hidden neurons would make more sense.
> >
> > Note that we do not comment on any upper bound on $k$. Our point only concerns the pattern in the number of hidden neurons, not on the number itself.
> >
> > **Q6.** Shouldn't $f^{PZ}_2$ be asymptotically included in $f^{PZ}_1$?
> >
> > **Response:** Yes it should, and we thank the reviewer for pointing out this error! We have corrected this in the revised manuscript.
> >
> > **Q7.** I am not sure I understood the sentence "any patters in G-SNN architecture space are unclear". Could you clarify or elaborate further on it?
> >
> > **Response:** We see now that our wording may have been confusing, and we have revised it.
> >
> > Our meaning, in detail, was the following:
> >
> > According to Thm. 4a, we enumerate the irreducible $G$-SNN architectures using only signed perm-reps satisfying a certain condition; signed-perm reps which do not satisfy this condition correspond to $G$-SNN architectures equivalent to those already enumerated. Now, imagine drawing a graph whose nodes are the conjugacy classes of subgroup pairs $(H, K)^G$. We would expect this graph to be closely related to the architecture spaces depicted, e.g., in Fig. 3. However, the nodes $(H, K)^G$ not satisfying the condition in Thm. 4a will have to be deleted, and while this can be done algorithmicly, we unfortunately do not have much intuition about what the resulting graph will look like in general. This makes it challenging to intuit what an architecture space of $G$-SNNs will look like apriori, unless we work it out manually for specific examples of $G$. Note, however, that this is only relevant for intuition; it does not hinder an algorithmic implementation.

---

### Official Review · Reviewer_7sGN · 2022-07-11

**Rating:** 6
**Confidence:** 2
**Soundness:** 3 good
**Presentation:** 3 good
**Contribution:** 3 good

**Summary:**

Before I begin, I have to note that I'm not very familiar with the concepts in algebraic topology required to fully understand this paper, so I find it unfortunate that this paper ended up in my batch. I have put my best efforts into evaluating this paper, but please take my words with a grain of salt.

This paper analyzes $G$-invariant 1-hidden-layer ReLU networks (called $G$-invariant SNNs or $G$-SNNs) for a finite orthogonal group $G$. Since fully-connected networks inherently contain redundancies such as permutation of hidden nodes, the authors first remove these redundancies via defining a "canonical parameterization" (Lemma 3). For any SNN $f$, the authors show that $f$ is $G$-invariant if and only if there exists a unique "signed permutation representation" $\rho: G \to PZ(n_*)$ such that the canonical parameterization satisfies a set of identities under $\rho$ (Lemma 4). Then it can be shown (Prop 18) that any $G$-invariant SNN can be decomposed into a sum of *irreducible* $G$-SNNs (whose corresponding $\rho$ is *irreducible*). Such irreducible $G$-SNNs has a particular form described in Theorem 5, and Theorem 1 & Prop 2 show that the corresponding $\rho$'s can be categorized into conjugacy classes of type 1 or 2, providing a "classification theorem" of $G$-SNNs.

**Questions:**

Q1. Although the prior works by Kondor and Trivedi (2018) and Cohen et al (2019) are for sure relevant, my understanding is that this paper is looking at a very slightly different problem. The existing two papers focus on $G$-equivariant networks (using $G$-equivariant layers) whereas this paper focuses on $G$-invariant networks. Is my understanding correct? Can your results say something as well about $G$-equivariant networks? In fact, if you sum up the output nodes of a $G$-equivariant network it gives you a $G$-invariant network, and the $a_* = a1$ part in Theorem 5b is reminiscent of this observation…

Q2. Can the theory extend to deeper networks? It seems to me that even defining the canonical parameterization is more challenging; what are other technical barriers? What about extensions to infinite groups (such as continuous rotation)?

Q3. In Lemma 3, how does $n_*$ compare with $n$ usually? Is it smaller or bigger? Or it depends on the parameters $a, W, b$?


**Limitations:**

I think the authors were clear about the limitations of their work, and they indicated some of them as their future research directions, which can be helpful for readers.

**Strengths And Weaknesses:**

Strengths

S1. This paper seems to convey solid mathematical theory that allows us to decompose $G$-invariant networks into several subnetworks of particular forms. This helps us understand the parametric forms of $G$-invariant networks better. Due to the lack of background, I was not able to check the correctness of the results though.

S2. Theorems 1 and 5 lead to an algorithm that can enumerate all irreducible $G$-SNN architectures for any finite orthogonal group $G$, which should prove helpful if you are a practitioner and you need to build a network that is invariant to a particular $G$.

S3. I found the connection to neural architecture search quite interesting, and I think further developing this theory can lead to a good algorithm for $G$-invariant NAS. Looking forward to the authors' future developments in this direction.

Weaknesses

W1. The paper is very math-heavy and I am worried if this paper would be readily accessible to the general audience at NeurIPS. The equations are quite heavy in notation and difficult to digest. Perhaps the authors can start with some illustrations (e.g., those in Sec 4) and then proceed to more abstract groups? Also consider reordering the main results a bit; personally I thought the paper would have been easier had it started from the canonical parameterization (Lemmas 3 and 4), then decomposition into irreducible $G$-SNNs, and then the classification of $G$-SNNs and corresponding irreducible signed perm-reps (Theorem 5, 1 and Prop 2).

W2. While the current theory seems concrete, I'm not sure yet about the potential impact of this paper because it is now limited to shallow networks. Extensions to deeper networks look quite challenging.

---

> ### Author Response · Authors · 2022-08-02
> **Response to reviewer 7sGN Part 1**
>
> We thank the reviewer for their helpful comments and suggestions. Below we try to respond to each point raised, one-by-one.
>
> **NOTE:** Please see revision/main.pdf in the supplementary material for the revised manuscript (there is content on page 10); the original manuscript is left unchanged. Key revisions are the following:
> 1. Rewrote Sec. 2.3 on group cohomology. Prop. 2 in the original submission is now in the supplementary material.
> 2. Lemma 3, Lemma 4, and Thm. 5 are now numbered Lemma 2, Lemma 3, and Thm. 4 respectively.
> 3. Added Thm. 5, which characterizes the network morphisms linking $G$-SNN architectures in architecture space.
> 4. Sec. 4.3 in the original submission is now Sec. 5.
>
> All numbered references to theorems, lemmas, etc. in our responses are based on our latest revision.
>
> ==========
>
> **W1.** The paper is very math-heavy and I am worried if this paper would be readily accessible to the general audience at NeurIPS. The equations are quite heavy in notation and difficult to digest. Perhaps the authors can start with some illustrations (e.g., those in Sec 4) and then proceed to more abstract groups? Also consider reordering the main results a bit; personally I thought the paper would have been easier had it started from the canonical parameterization (Lemmas 3 and 4), then decomposition into irreducible -SNNs, and then the classification of -SNNs and corresponding irreducible signed perm-reps (Theorem 5, 1 and Prop 2).
>
> **Response:** We certainly understand that our paper is quite dense, and we thank the reviewer for wading through the math to grasp the main points of the paper. We suspect the most opaque part of the paper is Sec. 2.3, the section on group cohomology. In hindsight, we see that the technical details of group cohomology (or even its definition) are not really necessary for most of the paper; its purpose is primarily to provide the basis for the directed graph visualizations in Fig. 1 as well as to connect our work to a fruitful area of mathematics.
>
> To simplify Sec. 2.3, we have moved Prop. 2 into the supplementary material and have completely rewritten Sec. 2.3 to give a high-level overview of group cohomology as it pertains to our paper.
>
> We had initially considered the paper organization that the reviewer suggests (starting with canonical parameterization, etc). While we could perhaps consider such an organization again given enough time, some care would have to be taken regarding the introduction of notation and concepts. For example, signed perm-reps would have to be discussed either right before or right after Lemma 3 (Lemma 4 in the original submission). Our hope with the current organization was that it would separate the "pure math" (Sec. 2) from the machine learning (Sec. 3), thereby streamlining both sections; and we hoped readers would be able to use Sec. 2 as a reference while reading Sec. 3. In short, we found there are pros and cons with both organizations.
>
> **W2.** While the current theory seems concrete, I'm not sure yet about the potential impact of this paper because it is now limited to shallow networks. Extensions to deeper networks look quite challenging.
>
> **Response:** Please see our response to Q2 below.

---

> > ### Author Response · Authors · 2022-08-02
> > **Response to reviewer 7sGN Part 2**
> >
> > **Q1.** Although the prior works by Kondor and Trivedi (2018) and Cohen et al (2019) are for sure relevant, my understanding is that this paper is looking at a very slightly different problem. [. . . ] Can your results say something as well about $G$-equivariant networks? In fact, if you sum up the output nodes of a $G$-equivariant network it gives you a $G$-invariant network, and the  part in Theorem 5b is reminiscent of this observation.
> >
> > **Response:** We thank the reviewer for this question, as we now see that our paper does have something to say about $G$-equivariant layers. In Lemma 3, Eq. 3 shows that the weight matrix of a $G$-SNN must be a $G$-equivariant map; i.e., the first layer of a $G$-SNN must be $G$-equivariant. Moreover, Eq. 4 shows that the second layer of a $G$-SNN must be invariant to a certain perm-rep of $G$ (please note that $G$-invariance is a special case of $G$-equivariance). Thus, in contrast to Kondor & Trivedi (2018) and Cohen et al (2019) who assume apriori that each layer of a $G$-invariant network prior to the activation is $G$-equivariant (with the last layer being $G$-invariant in particular), our work shows that this is necessarily the case for $G$-SNNs. Kondor & Trivedi (2018) even conjecture in their paper that the layers of a (deep) $G$-invariant network are necessarily $G$-equivariant; our work confirms this for the shallow case with ReLU activation. We note, however, that the case of deep networks may be more complicated, and the conjecture remains open.
> >
> > Finally, the distinction between $G$-invariance and $G$-equivariance can also be applied to the network as a whole-- i.e., to the output of the network with respect to its input. In our paper, we consider $G$-invariant networks as this is the natural first step. In future work, we may consider multi-output $G$-equivariant networks. \
> >
> > **Q2.** Can the theory extend to deeper networks? It seems to me that even defining the canonical parameterization is more challenging; what are other technical barriers? What about extensions to infinite groups (such as continuous rotation)? \
> >
> > **Response:** We are currently working on extending our results to deep networks, but we felt that we should not delay in sharing our results on shallow networks with the research community and that the deep result is beyond the scope of a first paper.
> >
> > As the reviewer rightly surmises, even a canonical parameterization for deep networks is challenging, but we have made significant progress and are hopeful that it can be done. Technically, induction on the depth of the network would be the natural approach, and we have indeed been able to classify some continuous redundancies in a deep network in this way. However, some of our equations involve couplings to deeper layers we have not reached in the induction, making them challenging to solve.
> >
> > There is also a result in the "identifiability" literature that the weights of a deep ReLU network can be determined up to certain redundancies if we exclude certain sets of measure 0. However, these 0-measure sets are complicated, and they cannot be ignored in the context of $G$-invariant networks as invariance can break even with arbitrarily small weight perturbations. Nevertheless, this literature gives us confidence that a deep generalization can be achieved if we take care to respect the subtleties of 0-measure sets involved.
> >
> > We would also like to comment that from what we have seen, it is not uncommon in theoretical deep learning research to first consider a simple tractable case, such as shallow networks, and then work to extend the results to more complicated settings later. For example, in the literature, this was the course taken in the study of infinitely wide and overparameterized neural networks.
> >
> > We have not yet considered infinite groups. We are not sure how continuous groups would interact with the ReLU activation, and groups such as continuous rotation groups may require us to consider alternative activation functions. Alternatively, we could work with large but finite discrete approximations to continuous groups, in which case our current theory should be applicable. \
> >
> > **Q3.** In Lemma 3, how does $n_*$ compare with $n$ usually? Is it smaller or bigger? Or it depends on the parameters $a, W, b$? \
> >
> > **Response:** $n_*$ is always smaller than $n$. In other words, $n_*$ is the smallest number of hidden neurons needed to express the function $f$ as an SNN. One source of redundancy (i.e., non-unique parameterizations) in SNNs is when two hidden ReLU neurons align to form a single linear term. In such a situation, the two hidden neurons are removed, and their sum is added to the $c_* x$ term. In this way, we can minimize the number of hidden neurons to $n_*$.
> >
> > We have revised the first paragraph of Sec. 3.1 and added some text immediately after Lemma 2 giving some insight into this.

---

### Public Comment · ~Celestine_Preetham_Lawrence1 · 2022-11-10
**Can this work be extended to shallow recurrent neural networks [1]?**

[1] Deep feedforward functionality by equilibrium-point control in a shallow recurrent network. https://openreview.net/forum?id=401LFvBGIb

Would also love to see your work put in perspective to the Group invariance theorem [2].

[2] Marvin, Minsky, and A. Papert Seymour. "Perceptrons." Cambridge, MA: MIT Press 6 (1969): 318-362.

---

### Meta-Review · Area_Chair_ACDF · 2022-09-05

**Recommendation:** Accept
**Confidence:** Less certain

**Metareview:**

Three reviewers gave quite positive ratings and comments on the paper. One another reviewer gave lower ratings. The AC thinks the reviewer with a low rating raised quite reasonable questions but found overall, the pros outweigh the cons (which seems to mostly stem from the unclarity in writing). The AC would recommend acceptance but also encourages the reviewers to incorporate the reviews and discussions in the final version.

**Award:**

No

---

### Decision · Program_Chairs · 2022-09-14

Accept